# Mutations primarily alter the inclusion of alternatively spliced exons

**Pablo Baeza-Centurion[1†], Belén Miñana[1†], Juan Valcárcel[1,2]\*, Ben Lehner[1,2,3]\***

[1]Centre for Genomic Regulation (CRG), The Barcelona Institute of Science and Technology (BIST), Barcelona, Spain; [2]Institució Catalana de Recerca i Estudis Avançats (ICREA), Barcelona, Spain; [3]Universitat Pompeu Fabra (UPF), Barcelona, Spain

**Abstract** Genetic analyses and systematic mutagenesis have revealed that synonymous, non-synonymous and intronic mutations frequently alter the inclusion levels of alternatively spliced exons, consistent with the concept that altered splicing might be a common mechanism by which mutations cause disease. However, most exons expressed in any cell are highly-included in mature mRNAs. Here, by performing deep mutagenesis of highly-included exons and by analysing the association between genome sequence variation and exon inclusion across the transcriptome, we report that mutations only very rarely alter the inclusion of highly-included exons. This is true for both exonic and intronic mutations as well as for perturbations in trans. Therefore, mutations that affect splicing are not evenly distributed across primary transcripts but are focussed in and around alternatively spliced exons with intermediate inclusion levels. These results provide a resource for prioritising synonymous and other variants as disease-causing mutations.

**\*For correspondence:**
juan.valcarcel@crg.eu (JVá);
ben.lehner@crg.eu (BL)

[†]These authors contributed equally to this work

## Introduction

Pre-mRNA splicing is a regulated step in eukaryotic gene expression in which introns are removed from primary transcripts and exons are joined together to form mature mRNAs that are subsequently exported to the cytoplasm for translation (*Braunschweig et al., 2013*). To carry out the splicing reaction, exon-intron boundaries in the pre-mRNA must be correctly identified. These boundaries consist of highly-conserved splice site sequences including intronic and exonic nucleotides. Apart from the splice sites, pre-mRNAs contain exonic and intronic sequences that interact with different splicing factors to regulate how often each exon is included in the final transcript (*Wang and Burge, 2008*). If this process is not properly defined, transcripts encoding altered amino acid sequences or premature stop codons can be produced. Depending on the location of the premature stop codon, this results in destruction of the mRNA by nonsense- mediated mRNA decay or production of a truncated protein (*Lindeboom et al., 2019*). mRNA degradation, truncated protein expression, abnormal protein sequences and altered isoform ratios can all disrupt cellular functions and cause disease (*Anna and Monika, 2018*; *Scotti and Swanson, 2016*; *Sterne-Weiler and Sanford, 2014*). Alterations in splicing contribute to diseases ranging from autoimmune diseases (*Agrebi et al., 2017*) to neurodevelopmental disorders (*Xiong et al., 2015*) and cancer (*Bonnal et al., 2020*; *Cancer Genome Atlas Research Network et al., 2018*; *Rahman et al., 2020*; *Rhine et al., 2018*; *Supek et al., 2014*). Indeed, in one analysis a third of disease-causing mutations were predicted to result in aberrant splicing (*Lim et al., 2011*) and in another 10% of 4964 disease-causing missense mutations altered the inclusion of cassette exons when tested in an episomal mini-gene construct (*Soemedi et al., 2017*).

To evaluate how often random mutations alter the inclusion of exons, several groups have recently subjected exons in mini-gene constructs to deep mutational scanning (DMS) (*Kinney and McCandlish, 2019*). In these DMS experiments, the effects of hundreds or thousands of mutations in

or around an exon are quantified in parallel by selection and sequencing. Exons subjected to DMS include *FAS* exon 6 (*Julien et al., 2016*), whose inclusion is altered in autoimmune lymphoproliferative syndrome (*Ben-Mustapha et al., 2018*); *RON* exon 11 (*Braun et al., 2018*), whose skipping can promote oncogenesis (*Collesi et al., 1996*); and *WT1* exon 5 (*Ke et al., 2018*), whose inclusion in leukemic cells confers resistance to chemotherapy drugs (*Yang et al., 2007*). These studies revealed that 60–70% of all single-nucleotide substitutions and mutations in over 90% of exon positions alter exon inclusion (*Braun et al., 2018*; *Julien et al., 2016*; *Ke et al., 2018*). One study where the flanking introns were also mutagenized (*Braun et al., 2018*) further revealed that mutations in ~80% of intronic positions flanking *RON* exon 11 also affect its inclusion.

However, these DMS experiments were performed on exons with intermediate or low inclusion levels: in the assayed conditions, wild-type (WT) *FAS* exon six had a percent spliced in (PSI, the percentage of mature transcripts that include the exon) of ~50%, *RON* exon 11 had a PSI of ~60%, and the different versions of *WT1* exon five had PSI values ranging between ~0 and ~70%. In contrast, in any given human tissue, most of the expressed exons are highly-included, for example ~60% of exons with a PSI > 10% actually have a PSI > 90% (*Figure 4— figure supplement 2A*). Notice that although a large fraction of exons in the human genome are included at very low levels (*Figure 4—figure supplement 2A*), these often correspond to pseudoexons or exons that are not usually expressed and may therefore be less physiologically relevant compared to other exons.

An alternative approach to random mutagenesis is to introduce a small number of mutations into an exon but in all possible combinations. For example, we recently quantified the effects on inclusion of all 3072 possible combinations of the 12 substitutions that accumulated in *FAS* exon six during its evolution from a constitutive exon in the last common ancestor of primates to an alternatively spliced exon in humans (*Baeza-Centurion et al., 2019*). This allowed us to quantify how the effect of each of these 12 mutations changes when it is made in different closely-related versions of exon six and revealed that the effects of mutations on exon inclusion 'scale' non-monotonically as the inclusion of an exon is increased. Specifically, as the PSI of an exon increases from 0 to 100%, the effect of a mutation increases to a maximum and then decreases again (*Figure 1A*), consistent with an underlying sigmoidal relationship between an additive trait affected by mutations and the inclusion level (*Baeza-Centurion et al., 2019*).

This scaling of mutational effects predicts that, although DMS experiments have revealed that many mutations have large effects on the inclusion levels of exons with intermediate PSI values, this may not be the case for exons with PSI values close to 100%. Here we first test this prediction by performing mutational scans of two exons with high inclusion levels. We find that, consistent with the scaling of mutational effects, mutations rarely substantially alter the inclusion of these exons. Analyses of additional mutagenesis datasets and human genetic variation strongly support this conclusion and show that exonic and intronic mutations – except those affecting the splice sites themselves – only rarely alter the splicing of exons with high PSI values. Mutations that alter exon inclusion are therefore very non-uniformly distributed across the human transcriptome and are focussed in and around alternative exons with intermediate inclusion levels. This provides an important framework for prioritising synonymous and other variants as candidate disease-causing mutations and for predicting from sequence when a change in splicing is – or is not – likely to be the causal molecular mechanism linking a genetic variant to a human disease.

## Results

### Mutational scaling predicts very few mutations will alter the splicing of highly-included exons

Our previous work indicated that the effects of mutations on exon inclusion can be small if the mutated exon is included at very high levels (*Baeza-Centurion et al., 2019*). In particular, the relationship between the effect of a mutation on splicing (that depends on the initial exon inclusion levels) and its biophysical effect (which is not dependent on the starting levels of inclusion) is given by an equation (derived in *Baeza-Centurion et al., 2019*):

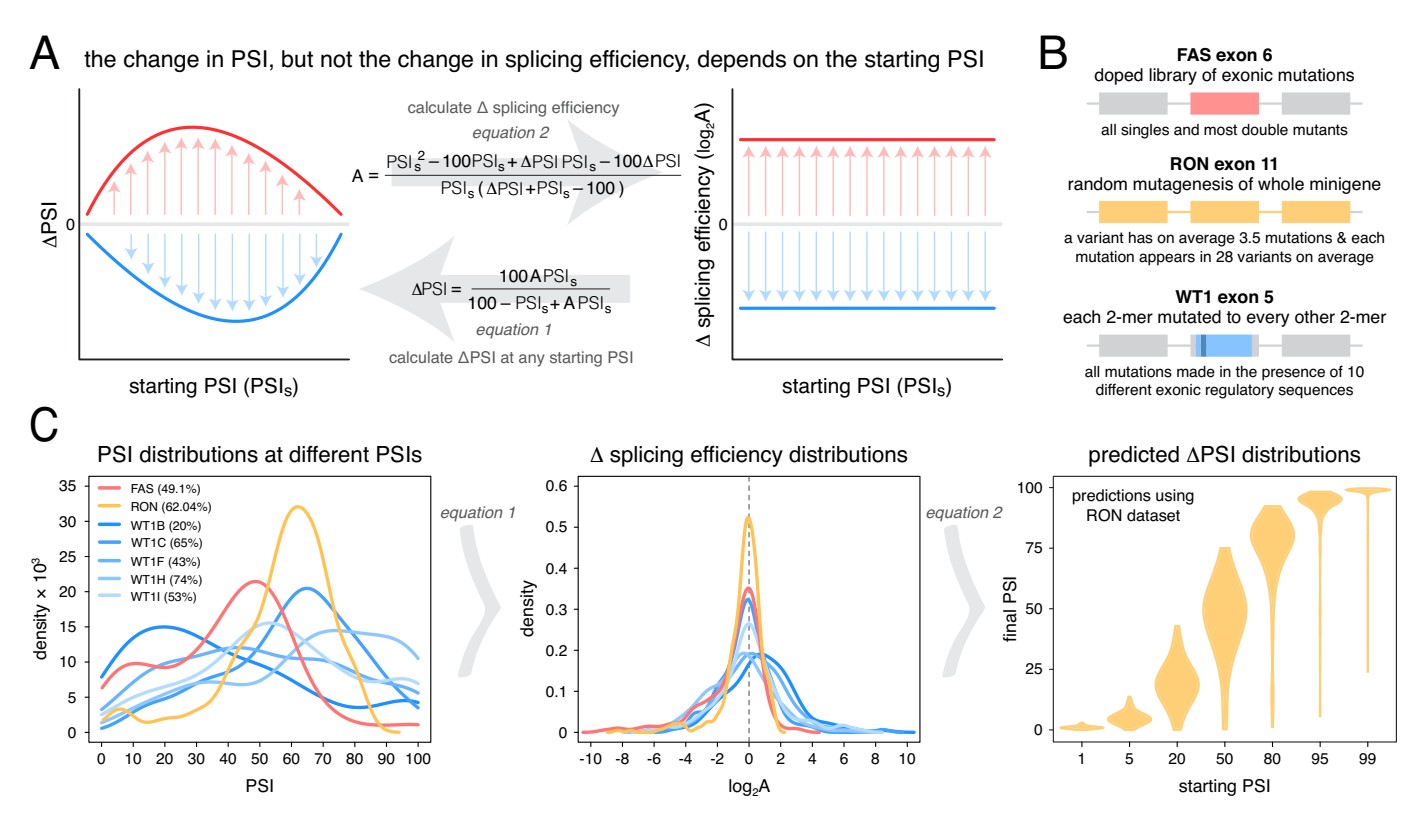

**Figure 1.** Scaling of mutation effects. (A) Mutation-induced changes in exon inclusion (ΔPSI) depend on the initial inclusion levels (starting PSI or PSI$_s$), but the underlying additive effect (A, which can also be interpreted as the change in splicing efficiency) of a mutation is independent of the starting PSI. The relationship between ΔPSI and A is given by precise mathematical equations and if the additive effect A of a mutation is known, its ΔPSI at any starting PSI can be calculated. (B) Previous deep mutagenesis experiments in three alternative exons. The *FAS* exon six experiment (*Julien et al., 2016*) involved systematically mutating all positions in the cassette exon. The *RON* exon 11 experiment (*Braun et al., 2018*) involved the mutagenesis of the entire minigene. The *WT1* exon five experiment (*Ke et al., 2018*) involved systematic mutagenesis of exonic positions except for splice sites. (C) The ΔPSI distributions generated by mutations in the experiments described in B can be converted to distributions of the underlying additive effects (using *Equation 1* shown in A). These distributions can be used to predict the distribution of ΔPSI values at any starting inclusion levels (using *Equation 2* shown in A), allowing us to scale a distribution of ΔPSI mutation effects for any starting PSI. The violin plots in the right-most panel show the PSI distributions from the *RON* exon 11 dataset (*Braun et al., 2018*) as they would look if the exon was included at the levels shown in the x axis. The online version of this article includes the following figure supplement(s) for figure 1:

**Figure supplement 1.** Predicted ΔPSI distributions using mutagenesis data from different alternative exons.

$$\Delta\text{PSI} = \frac{100 \cdot A \cdot \text{PSI}_s}{100 - \text{PSI}_s + A \cdot \text{PSI}_s} \tag{1}$$

where A is a parameter describing the change in splicing efficiency introduced by a mutation (its biophysical effect, equivalent -for example- to the loss or gain of binding affinity of one or more splicing regulatory factors), ΔPSI is the change in exon inclusion level caused by the mutation and PSIs is the starting PSI at which the ΔPSI is observed. Multiplying some of the terms by 100 ensures that mutation effects are calculated for an exon whose inclusion level can range from 0 to 100%. Therefore, this equation allows us to predict the PSI changes induced by a mutation at any starting PSI provided we already know its parameter A. Similarly, rearranging *Equation 1* and solving it for A allows us to estimate the biophysical effect of a mutation from ΔPSI measurements alone if we know the starting PSI at which those measurements were made (*Figure 1A*):

$$A = \frac{\text{PSI}_s^2 - 100 \cdot \text{PSI}_s + \Delta\text{PSI} \cdot \text{PSI}_s - 100 \cdot \Delta\text{PSI}}{\text{PSI}_s(\Delta\text{PSI} + \text{PSI}_s - 100)} \tag{2}$$

Unlike the ΔPSI effect of a mutation, the biophysical effect A of a mutation is always the same regardless of the inclusion levels of the exon (*Figure 1A*). We can therefore estimate A with measurements made at any given starting PSI (*Equation 2*). Once we know the value of A we can use it to predict and re-scale the mutation's ΔPSI when the affected exon is included at a different PSI level.

To illustrate this point with the effects of mutations on exon sequence configurations displaying high inclusion levels, we re-scaled (*Figure 1A*) the quantified effects of hundreds of mutations in three different alternatively spliced exons (*Figure 1B*). Deep mutagenesis of *FAS* exon 6 (*Julien et al., 2016*), *RON* exon 11 (*Braun et al., 2018*) and *WT1* exon 5 (*Ke et al., 2018* – we only used mutagenesis data from *WT1* exon five variants included at intermediate levels, between 20 and 70%) all revealed that many mutations have large effects on splicing (*Figure 1C*). However, when these mutational effects are rescaled for an exon sequence configuration displaying 99% PSI, the data from all three exons predict a very narrow distribution of changes in PSI (*Figure 1C*, *Figure 1—figure supplement 1*). For example, according to the distribution of mutational effects in *RON* exon 11, only 3.6% of exonic mutations are predicted to change the inclusion of an exon with 99% PSI by at least 10 PSI units (*Figure 1C*). This compares to 42.4% of mutations in an exon with a PSI of 50% (*Figure 1C*). A similarly narrow distribution of mutation effects is predicted if the exon were to be included at very low levels.

## Deep mutagenesis confirms mutations rarely have large effects in exons with high inclusion levels

To test the prediction that mutations generally have small effects on the inclusion of highly-included exons, we performed two new deep mutagenesis experiments (*Figure 2A–B*). First, we quantified the effects of all single-nucleotide changes in the inferred primate ancestor of *FAS* exon 6, which differs by 11 nucleotide substitutions from the human exon and is included in ~96% of mature transcripts in HEK293 cells (*Baeza-Centurion et al., 2019*). Second, we quantified the effects of all single-nucleotide changes in *PSMD14* exon 11, an exon that is highly included across all human tissues with this constitutive inclusion also highly conserved across mammalian species (*Tapial et al., 2017*). *PSMD14* encodes a subunit of the 26S proteasome with exon 11 included at ~98% in HEK293 cells (*Figure 2A*).

Although mutations have a strong effect in FAS exon 6, which is included at intermediate levels (~49%, *Figure 2A*), mutations had a much weaker effect on the highly-included exons (*Figure 2C*). Indeed, the PSI distributions in both libraries are narrow, with a mode near the WT levels of inclusion and 91% (ancestral *FAS* exon 6) and 86% (*PSMD14* exon 11) of mutations having an absolute effect <10 PSI units (*Figure 2C*). These distributions of mutational effects were accurately predicted by rescaling the distributions of the *FAS* exon 6, *RON* exon 11 and *WT1* exon five datasets (99% confidence intervals taking into account all of these datasets shown in *Figure 2E*; note that we excluded *WT1* exon five version B from this analysis, since it systematically predicted PSI distributions that were higher than expected that are not centered around the WT exon inclusion levels). The shape of the PSI distributions in both our experiments was well approximated by the rescaled distributions (note that we are not predicting the effect of each individual mutation). This shows that the distribution of mutational effects in an exon can be reasonably well predicted with data obtained from a completely unrelated exon. To further confirm this, we used the same approach to predict the ΔPSI distribution for single mutants in *SMN1* exon 7 (WT PSI = 99.4%; *Souček et al., 2019*). Mutations in this exon had very small effects, leading to a narrow distribution of effect sizes that was also accurately predicted (*Figure 2D,F*).

Thus, knowing the distribution of mutational effects in an example exon, we can successfully predict the distribution in other exons, provided that we know their starting inclusion levels. Moreover, these deep mutational scans confirm that mutations in highly-included exons generally have much smaller effects on inclusion than mutations in alternatively spliced exons with intermediate inclusion.

## Exonic mutations primarily alter the inclusion of exons with intermediate levels of inclusion

We next tested whether these conclusions, derived from the deep mutagenesis of three highly-included exons and three exons with intermediate inclusion, also apply to other exons in the genome. We first used data from two different multiplexed assays in which the effects of a small

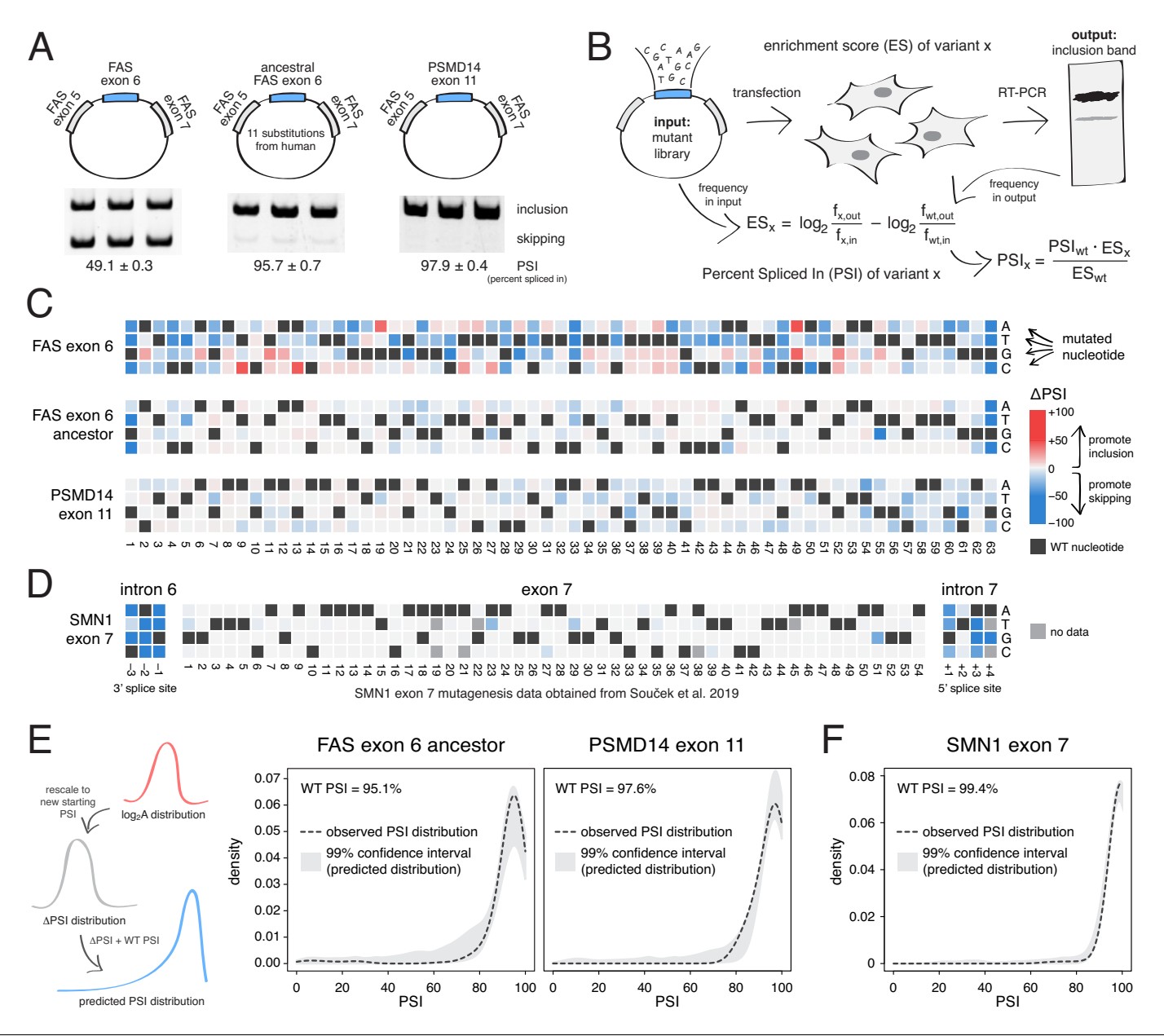

**Figure 2.** Deep mutagenesis of highly-included exons. (**A**) The inclusion levels of *FAS* exon 6 (gel image adapted from *Julien et al., 2016*), the ancestral *FAS* exon 6, and *PSMD14* exon 11. All inclusion levels were measured in HEK293 cells. (**B**) Experimental setup. (**C**) Heatmaps showing the inclusion levels of all single-nucleotide substitutions in *FAS* exon 6, the ancestral *FAS* exon six and *PSMD14* exon 11. (**D**) Heatmap showing the effects of mutations in *SMN1* exon 7 (*Souček et al., 2019*). (**E**) The PSI distribution of single-nucleotide substitutions in two exons included at 95.1% (the ancestral *FAS* exon 6) and 97.6% (*PSMD14* exon 11) was predicted using the rescaled distribution of mutation effects in *FAS* exon 6, *RON* exon 11 and *WT1* exon 5 (99% confidence band for the predicted distribution shown in grey). (**F**) The distribution of mutations in *SMN1* exon 7, included at 99.4% (*Souček et al., 2019*). The 99% confidence band for the predicted distribution is shown in grey.

The online version of this article includes the following figure supplement(s) for figure 2:

**Figure supplement 1.** Experimental validation of PSI values determined in our DMS experiments.

number of mutations were quantified in a mix of constitutive and alternative exons included at different levels (*Figure 3A*). In the first study, Adamson et al. built a plasmid library (the Vex-seq library) containing 2059 intronic and exonic variants from the ExAC database (*Exome Aggregation Consortium et al., 2016*) affecting 110 different exons (variants evenly distributed across exonic and

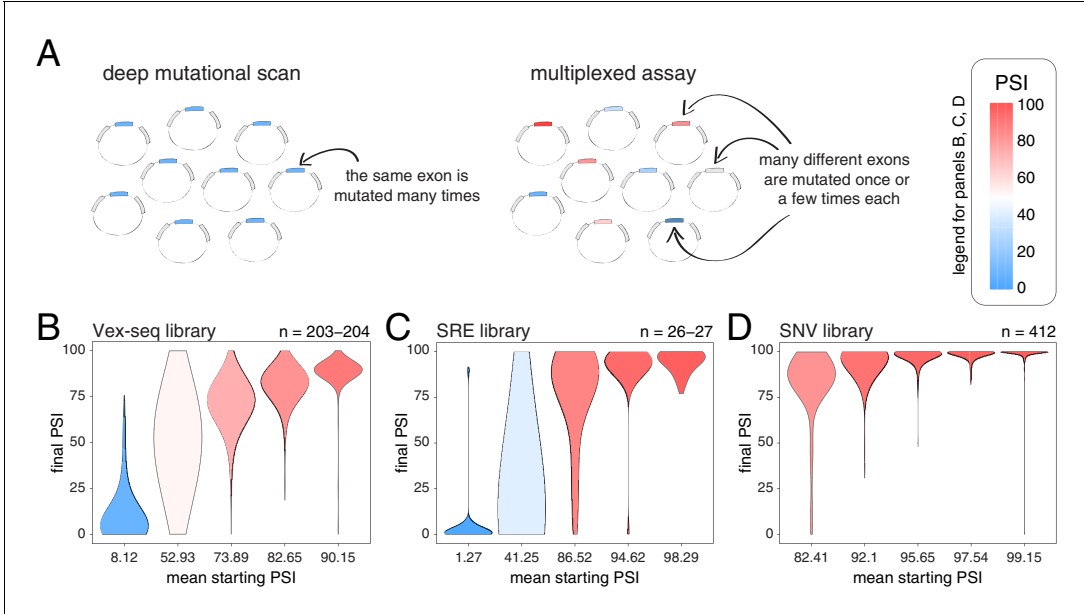

**Figure 3.** Across thousands of exons, exonic mutations have a stronger effect on the inclusion of exons with intermediate inclusion levels. (**A**) Cartoon highlighting the major difference between a deep mutagenesis experiment and a multiplexed experiment. Deep mutagenesis assays involve the analysis of many different mutations in the same exon. Multiplexed assays introduce one or a few mutations in many different exons. (**B**) Distribution of exonic mutation effects in the Vex-seq library, separated into five equally-populated bins according to the inclusion levels of the mutated exon. (**C**) Distribution of exonic mutation effects in the SRE library, separated into five equally-populated bins according to the inclusion levels of the mutated exon. (**D**) Distribution of exonic mutation effects in the SNV library, separated into five equally-populated bins according to the inclusion levels of the mutated exon.

The online version of this article includes the following figure supplement(s) for figure 3:

**Figure supplement 1.** Distribution of mutation effects in different multiplexed libraries.

**Figure supplement 2.** Effects of exonic mutations in the multiplexed datasets, binned by the starting PSI.

intronic positions, with a distribution of mutation effects centered around 0; *Figure 3—figure supplement 1*; *Adamson et al., 2018*). In the second study, Cheung et al. built two different libraries integrated into a specific site in the genome of HEK293 cell lines. The first library (SRE library) contained 6713 complex intronic and exonic mutations (involving the simultaneous substitutions of multiple nucleotides at once) designed to perturb splicing regulatory elements in 205 randomly selected exons and their surrounding intronic sequences (*Cheung et al., 2019*). The second library (SNV library) contains 27,733 intronic and exonic variants from the ExAC database across 2198 different exons.

We calculated the effects of all exonic mutations on exon inclusion in each of these three libraries and binned the data into five equally-populated bins according to the starting PSI (the exon inclusion levels in the absence of mutations). In the bins with exons included at intermediate levels, 55.2% (Vex-seq library), 69.2% (SRE library) and 36.2% (SNV library) of mutations changed inclusion levels by more than 10 PSI units (median absolute ΔPSI = 12.3, 18.3 and 7.0). In contrast and consistent with the scaling law and the DMS datasets, the distribution of mutational effects in bins containing the most highly-included exons was consistently narrow (*Figure 3B–D*, *Figure 3—figure supplement 2*), with 2.5, 14.8, and 0.5% of mutations changing inclusion by over 10 PSI units in the bins containing the most highly-included exons (median absolute ΔPSI = 2.1, 1.7 and 0.7, respectively). The larger effects of mutations in the SRE library is consistent with the more complex nature of these mutations which were designed to disrupt entire sequence motifs important for the splicing reaction, such as splice sites and splicing factor binding sites.

Mutations also had very small effects in exons that are mostly skipped in the two libraries containing exons with low inclusion levels (the Vex-seq and SRE libraries, *Figure 3B–C*, *Figure 3—figure supplement 2*), which is also highly consistent with the scaling of mutational effects. Thus, analysing

thousands of mutations in thousands of exons further confirms that whereas mutations frequently alter the inclusion of exons with intermediate inclusion levels, they rarely do so for highly-included exons.

## Common alternative alleles also primarily alter the inclusion of exons with intermediate levels of inclusion

Next, we used human genetic variation to evaluate the effects of mutations in their native genetic contexts. Using data from the GTEx consortium (*GTEx Consortium, 2017*) we tested the association between common exonic alternative alleles and exon inclusion levels across 635 humans and 27 tissues. For each allele, we used a linear model to estimate the ΔPSI associated with homozygosity (two copies) of the allele in each tissue. We split the estimated mutation effects into 25 equally populated bins according to the starting PSI (exon inclusion levels in the absence of an allele). As observed in the mutant libraries, common alternative alleles are associated with the largest PSI changes in exons with intermediate inclusion levels (data for heart tissue is shown in *Figure 4A*; data for all tissues is shown in *Figure 4—figure supplement 1*). 58.6% of alternative alleles in the 13[th] bin, corresponding to exons included at intermediate levels, had an absolute effect >10 PSI units. In

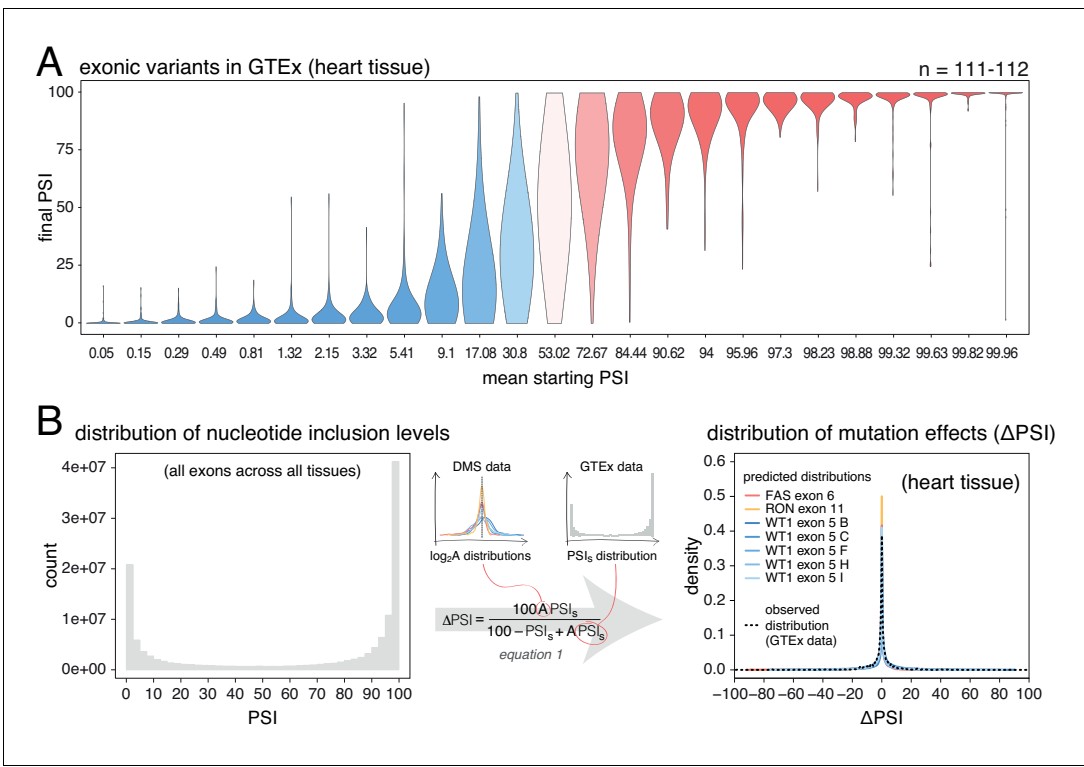

**Figure 4.** Common alternative alleles have a stronger effect on the inclusion of exons with intermediate inclusion levels. (**A**) Distribution of exonic allele effects in all exons and all tissues in the GTEx population. The data was split into 25 equally-populated bins according to the inclusion levels of each exon in the absence of the allele. (**B**) A random exonic mutation is unlikely to have an effect on splicing. Left. Bimodal distribution of exonic nucleotide inclusion levels for all exons across all tissues in the GTEx dataset. Right. The distribution of splice-altering effects of mutations in all human exons in heart tissue was predicted using the different alternative exon datasets as in *Figure 2* (coloured lines). The observed distribution of common allele-associated splicing changes is overlaid (dashed black line).

The online version of this article includes the following figure supplement(s) for figure 4:

**Figure supplement 1.** Effects of common exonic alternative alleles in different human tissues, binned by the starting PSI.

**Figure supplement 2.** Distribution of exon inclusion levels in the human genome.

**Figure supplement 3.** Distribution of genome-wide nucleotide inclusion levels.

**Figure supplement 4.** The distribution of splice-altering effects of mutations in all human exons, divided by tissue.

contrast and in agreement with the mutagenesis experiments, only 2.5% of alternative alleles had this effect in the 25th bin, which contains the most highly-included exons in the dataset. Mutations also had a similarly small effect in the 1st bin, containing the least included exons in the dataset.

Therefore, deep mutagenesis of individual exons (*Figure 2*), individual mutations in thousands of exons (*Figure 3*) and common alternative alleles (*Figure 4A*) all support the predictions of the empirical scaling law that random exonic mutations rarely alter the inclusion of highly-included exons but frequently alter the inclusion of alternatively spliced exons with intermediate inclusion levels.

## Estimating the distribution of exonic mutational effects for the human genome

We next used the DMS data to predict the distribution of mutational effects (ΔPSIs) for all single-nucleotide mutations in all human exons. Here the goal is not to predict the actual effect of each individual mutation, but rather to predict the distribution of changes in inclusion for all of the possible point mutations in each exon. We can then combine these distributions to obtain an estimate of the distribution of effects for all possible mutations in all exons in the human genome in each tissue. To do this, we rescaled the distribution of mutational effects from the alternative exon DMS datasets (*Figure 1C*) to the inclusion level of every exon in every tissue, resulting in a ΔPSI distribution for mutations in each exon (and each tissue) in the human genome. Then, to obtain the genome-wide distribution of mutational effects on exon inclusion, we combined these distributions, normalising by the length of each exon.

Human genome-wide exon inclusion levels are bimodal, with few exons in any given tissue included at intermediate levels and most exons almost completely included or skipped (*Figure 4—figure supplement 2*; similar results obtained after normalising by exon length, *Figure 4B Figure 4—figure supplement 3*). Combining data across 30 GTEx tissues, only 28% of exons have an intermediate PSI between 10% and 90% (*Figure 4B*, *Figure 4—figure supplement 2*, *Figure 4—figure supplement 3*). In these exons, 43% of mutations are predicted to affect splicing by more than 10 PSI units. However, since the majority (72%) of exons have inclusion levels < 10% or >90%, the distributions of mutational effects from the DMS datasets predict that most exonic mutations have zero or only very small effects on inclusion in any tissue (*Figure 4C*). Indeed, in each tissue the predicted distribution of mutational effects is unimodal and centered at zero, with 81% (95% confidence interval 76.6–86.8) of all exonic mutations predicted to alter inclusion by <10 PSI units (79.5%–88.4% of exonic mutations, depending on the tissue, *Figure 4—figure supplement 4*).

These predictions agree well with the estimated effects of common exonic alternative alleles (GTEx data, black dashed line in *Figure 4B*, *Figure 4—figure supplement 4*), only 14–17% of which (depending on the tissue) are associated with changes in inclusion greater than 10%. Therefore, with the exception of mutations in exons with intermediate PSI, the vast majority of single-nucleotide changes in human exons are expected to have zero or very small effects on exon inclusion.

## Intronic mutations also rarely alter the inclusion of highly-included exons

Mutations in introns can also alter the inclusion of neighbouring exons (*Wang and Burge, 2008*). To-date, only a single DMS study has quantified the effects of a large number of intronic mutations on exon inclusion (*Braun et al., 2018*). In total, Braun et al. tested the impact of 502 single-nucleotide substitutions in the short introns surrounding *RON* exon 11 on the inclusion levels of this exon, finding a similar distribution of mutational effects as for exonic mutations (*Figure 5A*). Re-scaling these mutational effects predicts that 30% of flanking intronic mutations will alter the inclusion level of a 50% included exon by >10 PSI units. This is in contrast to only 6.4% of mutations in an exon included at 99%.

Analyses of intronic mutations in three multiplexed libraries support this conclusion (*Figure 5B–D*, *Figure 5—figure supplement 1*). In the bin with the 20% of exons included at the most intermediate levels, 35.6% (Vex-seq library), 51.9% (SRE library), and 33.5% (SNV library) of mutations altered inclusion by >10 PSI units (median absolute ΔPSI = 6.5, 10.0 and 5.9). In comparison, only 1.84, 11.1, and 1.44% of mutations had this effect in the bins containing the most highly-included exons (20% most included exons in each dataset – median absolute ΔPSI = 2.1, 2.3, and 0.74, respectively).

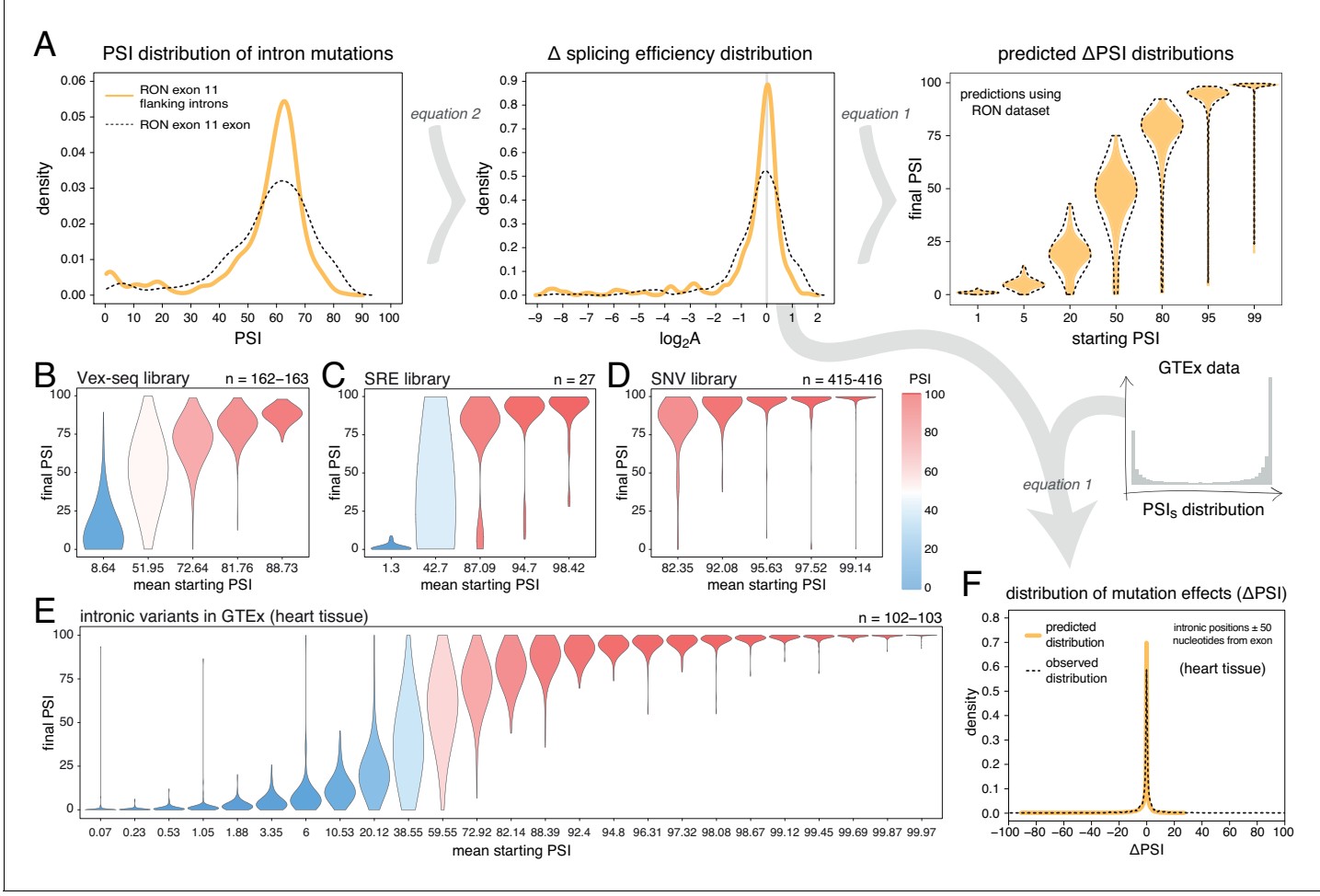

**Figure 5.** Intronic mutations have a stronger effect on the inclusion of exons with intermediate inclusion levels. (A) The distribution of intronic mutation effects in the *RON* exon 11 dataset can be converted into a distribution of effects on splicing efficiency, which can, in turn, be used to predict the distribution of mutation effects at any starting PSI (similar to *Figure 1C*). The black dashed lines show the corresponding distributions for exonic mutations. (B) Distribution of intronic mutation effects in the Vex-seq library, separated into five equally-populated bins according to the inclusion levels of the affected exon. (C) Distribution of intronic mutation effects in the SRE library, separated into five equally-populated bins according to the inclusion levels of the affected exon. (D) Distribution of intronic mutation effects in the SNV library, separated into five equally-populated bins according to the inclusion levels of the affected exon. (E) Distribution of intronic alternative allele effects in introns flanking all exons in heart tissue (GTEx data). The data was split into 25 equally-populated bins according to the inclusion levels of each exon in the absence of the allele. (F) The distribution of mutation effects in the flanking introns of *RON* exon 11, in combination with the distribution of nucleotide inclusion levels, can be used to predict the distribution of genome-wide flanking intronic mutation effects (yellow line). The black dashed line shows the effects distribution observed for common intronic alternative alleles in the GTEx population (heart tissue).

The online version of this article includes the following figure supplement(s) for figure 5:

**Figure supplement 1.** Effects of intronic mutations in the multiplexed datasets, binned by the starting PSI.

**Figure supplement 2.** Effects of common intronic alternative alleles in different human tissues, binned by the starting PSI.

**Figure supplement 3.** Splice-altering effects of changing the flanking introns, binned by the initial inclusion levels.

**Figure supplement 4.** Genome-wide distribution of splice-altering effects of common intronic alternative alleles.

We next turned to human genomic variation to evaluate whether intronic mutations in their native genomic context also have stronger effects on the splicing of intermediately-included exons. Considering alternative alleles within 50 nucleotides of each exon and all exons in all tissues, 26.5% of common intronic alleles near the exons with the most intermediate inclusion levels are associated with a change in inclusion of >10 PSI units (bin 13 out of 25; median absolute ΔPSI = 4.9), compared to only 1.6% in the most highly-included exons (top bin of 25, median absolute ΔPSI = 0.02; *Figure 5E*, data split by tissue shown in *Figure 5—figure supplement 2*). Similarly small effects were observed

for the bin containing the least included exons in the genome. Human genetic variation is therefore also consistent with intronic alternative alleles rarely affecting the inclusion of highly (and lowly) included exons.

We also considered a more complex sequence alteration – the complete exchange of one intron for another. The SRE library was built twice with different flanking introns (*Cheung et al., 2019*). Since changing an intron could be considered an extreme case of intronic mutation, we compared the inclusion levels of every allele in this library between the two intron contexts. In agreement with the data for simpler mutations, the effect of changing the flanking introns was strongest in exons included at intermediate levels (*Figure 5—figure supplement 3*).

Finally, we used the distribution of intronic mutational effects in the *RON* exon 11 dataset in combination with the PSI values of all human exons, to predict the genome-wide distributions of effects for intronic mutations within 50 nucleotides of each exon in each human tissue (pooled data for all exons in all tissues shown in *Figure 5F*, data split by tissue shown in *Figure 5—figure supplement 4*). The predicted distributions were similar to the ΔPSI distribution of common intronic alternative alleles: most mutations have little or no impact on exon inclusion and 8.7% of all proximal intronic mutations are predicted to change exon inclusion by more than 10 PSI units, compared to 7–13% of common alternative alleles across human tissues analysed by GTEx. However, focussing on intronic alternative alleles near exons whose inclusion levels range between 10 and 90% reveals that 21–31% (depending on the tissue) of common alternative alleles near these exons affect splicing by at least 10 PSI units.

Therefore, proximal intronic and exonic mutations have a similar effect on inclusion, with most proximal intronic mutations having very small or no effects on the inclusion of neighbouring exons except for the subset of exons with intermediate inclusion levels.

## Changes in trans also have stronger effects on the inclusion of exons with intermediate inclusion levels

The inclusion levels of exons can be affected by proximal mutations in cis but also by perturbations in trans, for example mutations or changes in the expression of other genes. To compare how a complex *trans* perturbation alters the inclusion of many different exons, we quantified how the inclusion level of every genotype of every mini-gene construct in the Vex-seq library changes between two cell types. Consistent with our results for the effects of *cis* mutations, the effect of changing the cell type was strongest in exons included at intermediate levels. Minimal effects were observed in highly and lowly included exons (*Figure 6*). Analyses of the effects of changes in inclusion between human tissues are consistent with this (for example, between skin and brain tissues, *Baeza-Centurion et al., 2019*).

## Mutations have stronger effects in highly-included alternatively spliced exons than in constitutive exons

Since alternative exons are often included at intermediate levels, our results so far suggest these exons are particularly susceptible to the splice-altering effects of mutations. However, alternative exons are sometimes included at very high levels (depending on the tissue). We therefore asked if mutations have a stronger effect on inclusion in alternative exons (exons that are sometimes but not always highly-included) than in constitutive exons (exons that are always highly-included across cell types) under conditions in which both types of exon are included at similar levels. To test this, we defined constitutive exons as those with a PSI of at least 90% in all 30 GTEx human tissues, and alternative exons as those with a PSI under 90% in at least one tissue.

Candidate skipping-promoting alleles (alternative alleles in the GTEx population associated with lower levels of exon inclusion) have stronger effects on the inclusion of alternative exons. At the same starting PSI, these alternative alleles are associated with a decrease in the inclusion of an alternative exon by ~5 PSI units more than in a constitutive exon (t test p-value<2.2e−16 for exonic alternative alleles, t test p-value < 2.2e−16 for intronic alternative alleles – *Figure 7A*, *Figure 7—figure supplement 1*, *Figure 7—figure supplement 2*). This distinction between the effects of alternative alleles in constitutive and alternative exons is observed regardless of the thresholds used to define these two classes of exon, with the largest differences (up to 15 PSI units) observed when alternative exons are defined as those with a PSI < 60% in at least one tissue (*Figure 7—figure supplement 3*).

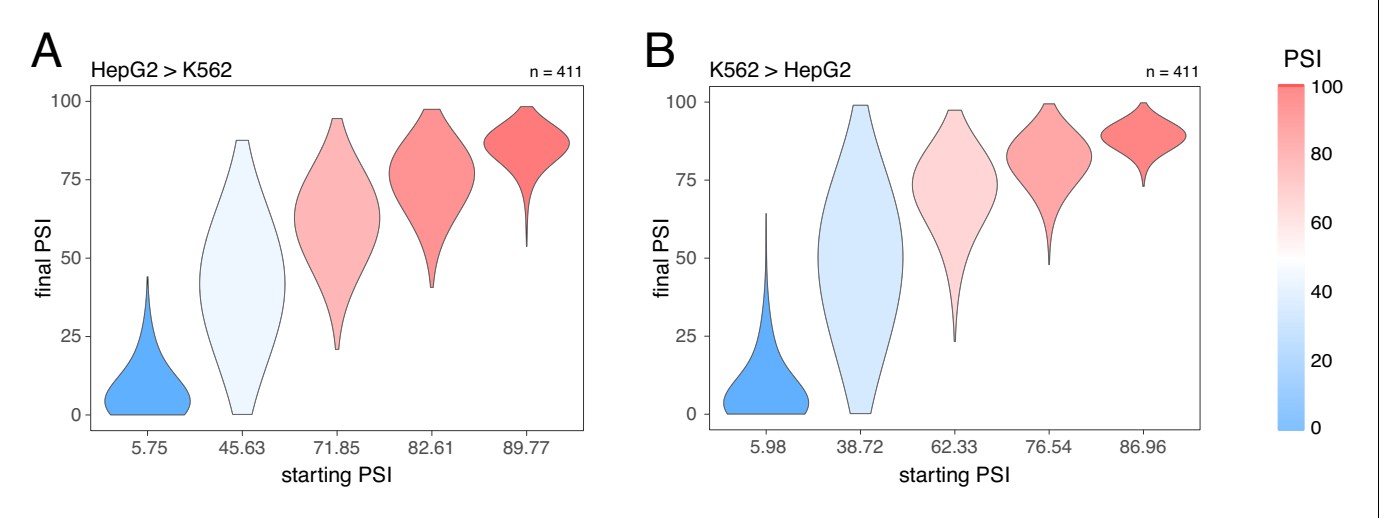

**Figure 6.** Splice-altering effects of a complex perturbation in trans binned by the initial inclusion levels. (A) The effect of moving from HepG2 cells (the initial PSI condition) to K562 cells (the final PSI condition). (B) The effect of moving from K562 cells (the initial PSI condition) to HepG2 cells (the final PSI condition). The numbers above the plots show the number of data points in each bin.

In subsequent analyses we therefore re-defined alternative exons as those with a PSI < 60% in at least one tissue.

Constitutive exons have been previously shown to contain a higher density of exonic splicing enhancers (ESEs) than alternative exons (*Ke et al., 2011*; *Wang et al., 2005*). ESEs are exonic sequences recognized and bound by splicing factor proteins that promote exon inclusion (*Blencowe, 2000*). We downloaded a list of 1182 putative ESE hexamer sequences from *Ke et al., 2011* and confirmed that constitutive exons in our dataset have a higher ESE density than alternative exons (*Figure 7B*). Limiting our analysis to highly-included alternative exons, we find that they are still enriched among exons with the fewest ESEs (by comparing constitutive exons against alternative exons with a PSI > 90% in at least one tissue, *Figure 7—figure supplement 4*; or by comparing constitutive exons in a given tissue against alternative exons with a PSI > 90% in that same tissue, *Figure 7—figure supplement 5*). These results hold true even if we fine-tune our analysis to include only those exons with very strong or very weak splice sites (*Figure 7—figure supplement 6*). Therefore, one reason why mutations have smaller effects in constitutive exons may be due to enhancer redundancy (*Figure 7C*). Similarly, exonic splicing silencers (ESSs) are enriched among alternative exons although ESSs are not found in alternative exons that are included with a PSI of 90% in at least one tissue (*Figure 7—figure supplement 7*). One possibility that explains this observation is that, while alternative exons included at intermediate levels make frequent use of ESSs to modulate their levels, those with high inclusion levels in some tissues establish their 'extreme' phenotypes (very high inclusion/high skipping) through other mechanisms.

We next compared, across both types of exon, the density of sequences that differ from any one ESE by only one nucleotide substitution (hereafter, 'suboptimal ESEs' – note that suboptimal ESEs also include ESE sequences that are one substitution away from each other). Alternative exons were enriched among exons with the smallest numbers of suboptimal ESEs, and this is also true after restricting our analysis to highly-included alternative exons (*Figure 7D*, *Figure 7—figure supplement 8*, *Figure 7—figure supplement 9*). Because suboptimal ESEs are one substitution away from an ESE, a mutation is more likely to create a new ESE sequence in a constitutive exon than in an alternative exon. This suggests two additional hypotheses that could explain why, at the same initial inclusion level, mutations have stronger effects in alternative exons (*Figure 7E*). First, ESEs in constitutive exons might be more robust to the effects of mutations than ESEs in alternative exons. Second, constitutive exons might contain many cryptic ESEs that could be activated by mutations that disrupt existing ESEs.

To test the first hypothesis, we split all 1182 ESE hexamers into different groups, depending on the number of suboptimal ESEs they contain (how many substitutions result in another ESE). In an

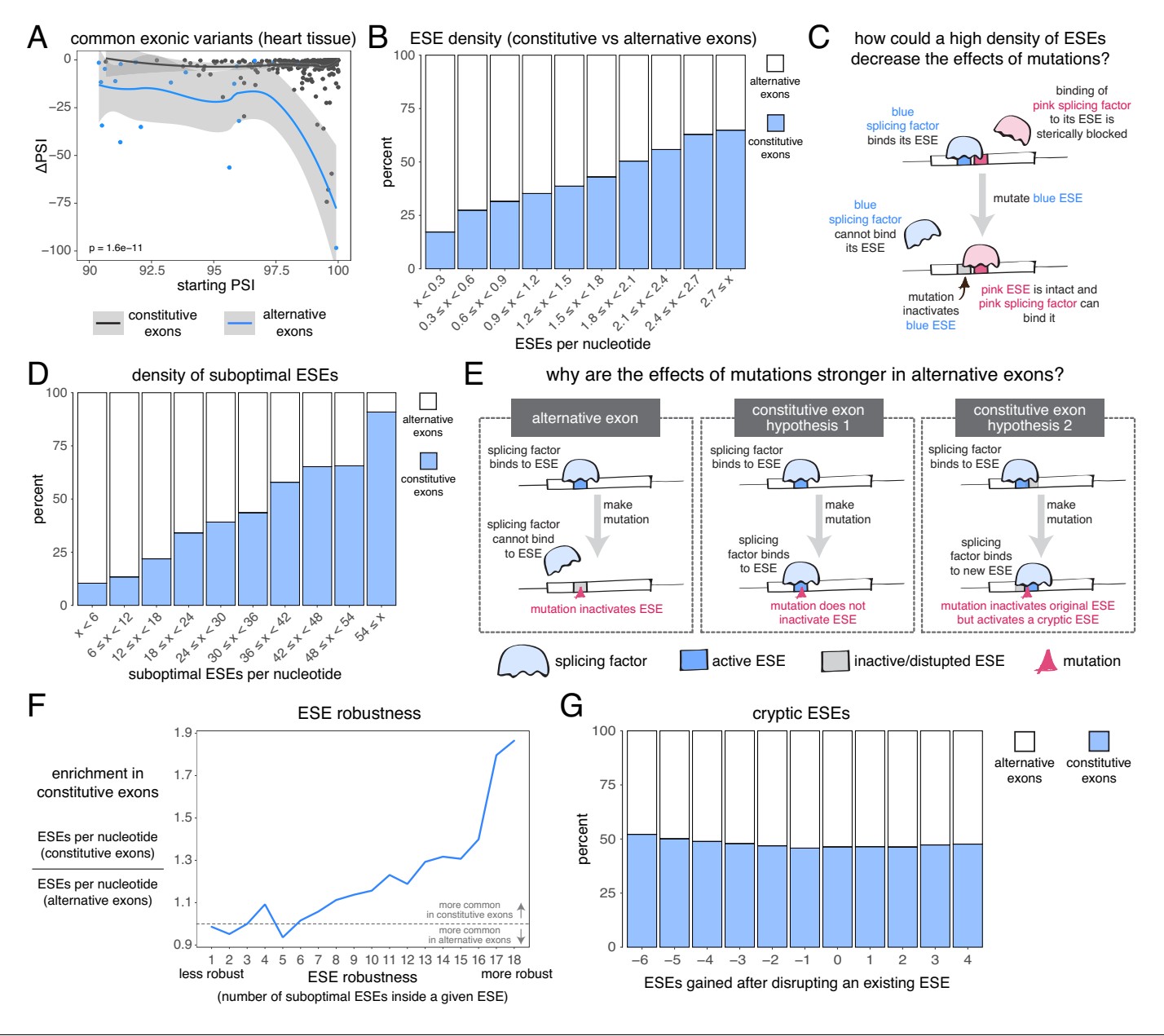

**Figure 7.** The effects of common alternative alleles in constitutive vs alternative exons. (**A**) At the same starting PSI, the effects of skipping-promoting exonic alleles in alternative exons (blue) are stronger than in constitutive exons (black). Data summarised with loess curves and 95% confidence bands. (**B**) All exons were binned into 10 groups depending on their density of exonic splicing enhancers (ESE). Constitutive exons (blue) were enriched in the bins with a higher density of ESE motifs, and alternative exons (white) were enriched in the bins with a lower ESE density. (**C**) A hypothetical mechanism illustrating how a high density of ESE motifs in an exon can result in sequence redundancy and decrease the effects of splicing mutations. (**D**) All exons were binned into 10 groups depending on their density of 'suboptimal ESE' sequences. Suboptimal ESEs were defined as sequences that differ from an ESE by one nucleotide substitution. Constitutive exons (blue) were enriched in the bins with a higher suboptimal ESE density, and alternative exons (white) were enriched in the bins with a lower density of such motifs. (**E**) Two hypotheses for how a higher density of suboptimal ESEs can decrease the effects of mutations in constitutive exons. Left: a mutation in the ESE of an alternative exon disrupts the ESE, leading to lower levels of inclusion. Middle: a mutation in the ESE of a constitutive exon might not disrupt the ESE and the mutation has little to no effect on splicing. Right: a mutation in the ESE of a constitutive exon might disrupt the ESE, but might activate another ESE instead, compensating for the loss of the original ESE. (**F**) ESEs that are robust to the effects of mutations are more common in constitutive exons. ESE robustness was defined as the number of ESE sequences accessible, from an existing ESE, via one nucleotide substitution. This definition allowed us to group all ESE sequences into 18 groups ranging from 1 (least robust) to 18 (most robust). For each group, we calculated the average number of ESEs per nucleotide in constitutive and alternative exons, and calculated the relative enrichment of sequences in constitutive over alternative exons by taking the ratio of these two numbers. (**G**) Constitutive and

*Figure 7 continued on next page*

*Figure 7 continued*

alternative exons have similar numbers of cryptic enhancers. We counted the number of ESEs created upon disrupting each ESE in an exon with a point mutation (without allowing for the creation of ESEs that occupy the exact same six nucleotides as the disrupted ESE).

The online version of this article includes the following figure supplement(s) for figure 7:

**Figure supplement 1.** The effects of common exonic alternative alleles in constitutive (black) and alternative (blue) exons in different human tissues.
**Figure supplement 2.** The effects of common intronic alleles in constitutive (black) and alternative (blue) exons in different human tissues.
**Figure supplement 3.** Using different thresholds to define constitutive and alternative exons.
**Figure supplement 4.** Density of ESEs in constitutive and alternative exons (included at >90% in at least one tissue).
**Figure supplement 5.** Density of ESEs in constitutive and alternative exons with a PSI > 90% in each human tissue.
**Figure supplement 6.** Density of ESEs in constitutive and alternative exons, accounting for splice site strength.
**Figure supplement 7.** Density of ESS hexamers.
**Figure supplement 8.** Density of suboptimal ESEs in constitutive and alternative exons (included >90% in at least one tissue).
**Figure supplement 9.** Density of suboptimal ESEs in constitutive and alternative exons with a PSI > 90% in each human tissue.
**Figure supplement 10.** Enrichment of ESEs in constitutive exons vs. ESE robustness (considering only alternative exons with a PSI > 90% in at least one tissue), for all exons in all human tissues.
**Figure supplement 11.** Robustness of ESEs in constitutive exons vs. ESE robustness in each human tissue (considering only alternative exons with a PSI > 90% in that tissue).
**Figure supplement 12.** Robustness of ESEs in constitutive exons vs robustness of ESEs in alternative exons, accounting for splice site strength and exon inclusion levels.
**Figure supplement 13.** Nucleotide distance between consecutive ESEs.
**Figure supplement 14.** We counted the number of ESSs created upon disrupting each ESE in an exon with a point mutation (without allowing for the creation of ESEs that occupy the exact same six nucleotides as the disrupted ESE).

ESE that is 100% robust to single-nucleotide substitutions, every possible substitution produces another ESE hexamer. Conversely, in a hexamer that is susceptible to mutations, few substitutions should result in an ESE. Robust ESEs were over-represented in constitutive exons, with the most robust hexamers being almost twice as likely to be found in constitutive as in alternative exons (*Figure 7F*). This is true even after limiting our analysis to highly-included alternative exons (*Figure 7—figure supplement 10*, *Figure 7—figure supplement 11*). The least robust ESE hexamers are generally over-represented in alternative exons (*Figure 7F*, *Figure 7—figure supplement 10*, *Figure 7—figure supplement 11*), and this is especially the case among exons surrounded by strong splice sites (*Figure 7—figure supplement 12*). However, considering only those exons surrounded by weak splice sites, the least robust ESEs are also enriched among constitutive exons (*Figure 7—figure supplement 12*), suggesting that alternative exons contain weak ESEs preferentially in combination with strong splice sites.

To test the second hypothesis, we counted how many ESEs were gained or lost in every exon after introducing a point mutation in each of its ESEs (not allowing the mutations to create another ESE that spans the same six nucleotides). Constitutive exons were not more likely than alternative exons to create new ESEs after inactivating an existing ESE (*Figure 7G*). Instead, constitutive exons were slightly more likely to destroy multiple sites at once, suggesting the presence of overlapping motifs in these exons which can be simultaneously deactivated by a single point mutation. Indeed, an analysis of overlapping ESE motifs reveals that these occur slightly more commonly in constitutive than in alternative exons (*Figure 7—figure supplement 13*). However, alternative exons are not more likely than constitutive exons to create an ESS motif after inactivating an existing ESE (*Figure 7—figure supplement 14*).

Taken together, our results reveal that, even at the same inclusion level, mutations have a stronger effect on the splicing of alternative exons than on the splicing of constitutive exons, and that this could be due to the higher density of ESEs in constitutive exons, as well as ESEs in constitutive exons being more robust to the effects of mutations.

## Splice site mutations do frequently alter the inclusion of highly-included exons

Exon-intron boundaries are defined by splice site sequences that are recognized by spliceosomal components before initiating the splicing reaction. The first and last two positions of each intron are the most highly-conserved positions (*Yeo and Burge, 2004*) and mutating these 'invariant' dinucleotides has been extensively reported to disrupt splicing (*Anna and Monika, 2018*; *Adamson et al.,*

*2018*; *Cheung et al., 2019*; *Rogan et al., 1998*; *Scotti and Swanson, 2016*; *Sterne-Weiler and Sanford, 2014*). Consistent with the high conservation of the 'invariant' dinucleotide intron positions, all mutations at these positions are associated with large decreases in splicing efficiency in the *RON* exon 11 dataset (*Figure 8A*). Consistent with the first exon base being part of the consensus for U2AF1 binding (*Wu et al., 1999*) and the last exon base being often part of extended basepairing with U1 snRNA (*Roca et al., 2005*), all mutations in the first and last exon bases of *RON* exon 11 (*Figure 8A*) or the ancestral *FAS* exon 6 (*Figures 2C* and *8B*) had strong skipping-promoting effects (e.g. 100% of mutations in the first or last positions of the ancestral *FAS* exon six had an absolute effect greater than 10 PSI units, compared with 87.4% in the rest of the exon). These results suggest that mutating these positions might alter the inclusion of even very highly-included exons, although we did not observe these results in the *PSMD14* exon 11 dataset (*Figure 2C*, *Figure 8B*). This suggests that splice site mutations display stronger effects in highly-included alternative exons than in constitutive exons.

To test the extent to which splice site mutations can affect the inclusion of highly-included exons, we analysed the three multiplexed libraries described above. Regardless of the starting PSI, mutations in the 'invariant' dinucleotides had stronger effects than mutations elsewhere in an intron (100%, 53.8% and 80% of invariant dinucleotide positions in the bin with the highest-included exons in the Vex-seq, SRE and SNV multiplexed assays, respectively, had an effect greater than 10 PSI units compared with 1.8%, 11.1% and 1.4% in the rest of the intron, *Figure 8C–E*). Similarly, mutating the first or last exonic bases had a stronger effect than mutating any other exonic base. 36.1% (SNV dataset) of substitutions in the last position and 9.5% of substitutions in the first exonic position in the bin with the highest-included exons altered splicing by more than 10 PSI units compared with 0.5% elsewhere in the exon (*Figure 8F–G*).

These results are in agreement with many previously-published studies (*Cheung et al., 2019*; *GTEx Consortium et al., 2015*; *Zhang et al., 2018*), and are consistent with the last exonic base having a greater information content in the donor splice site motif than the first exonic base has in the acceptor splice site motif (*Yeo and Burge, 2004*).

## Discussion

Systematic mutagenesis of three different alternatively spliced exons with intermediate inclusion levels has revealed that mutations at most positions in exons can have substantial effects on their inclusion levels (*Braun et al., 2018*; *Julien et al., 2016*; *Ke et al., 2018*). Considered in isolation, this finding might suggest that changes in splicing could be one of the major mechanisms by which exonic mutations cause phenotypic variation and human disease. However, the effects of mutations on inclusion levels have recently been proposed to scale with the level of inclusion, predicting mutations will only rarely have large effects on the inclusion levels of highly-included exons (*Baeza-Centurion et al., 2019*). Since highly-included exons make up most of the expressed exons in the genome in any cell type (*Figure 4B* and *Chen, 2014*), this leads to the prediction that exonic mutations will actually only rarely change exon inclusion and that mutations having large effects on splicing correspond to the minority of exons included at intermediate levels in a cell type.

To test these predictions, we performed deep mutagenesis of two highly-included exons. With the exception of mutations in canonical splice sites, we found that mutations generally had little influence on the inclusion of highly-included exons. Deep mutagenesis of a third exon supports this conclusion (*Souček et al., 2019*) as do the effects of thousands of mutations in libraries spanning thousands of different exons (*Figure 3B–D*) and the associations between common human genetic variation and changes in inclusion (*Figure 4A*). Analyses of mutations in introns suggest that the same conclusion applies to intronic mutations (*Figure 5B–E*) and further analyses suggest that it is also true for *trans* perturbations (*Figure 6*). The vast majority of single-nucleotide mutations in human exons and introns should therefore be expected to have negligible effects on exon inclusion.

Our conclusions may appear to be in conflict with previous reports of exonic splicing enhancers (exonic sequences that promote higher inclusion) in highly included exons. However, we do not think this is actually the case. In these studies, enhancers were often identified by deleting or substituting entire exonic domains (*Liu et al., 1998*; *Watakabe et al., 1993*) or by using antisense oligonucleotides to block binding to an entire domain (*Aartsma-Rus et al., 2006*). Moreover, studies reporting strong effects of single-nucleotide changes generally involved in vitro experiments in which the

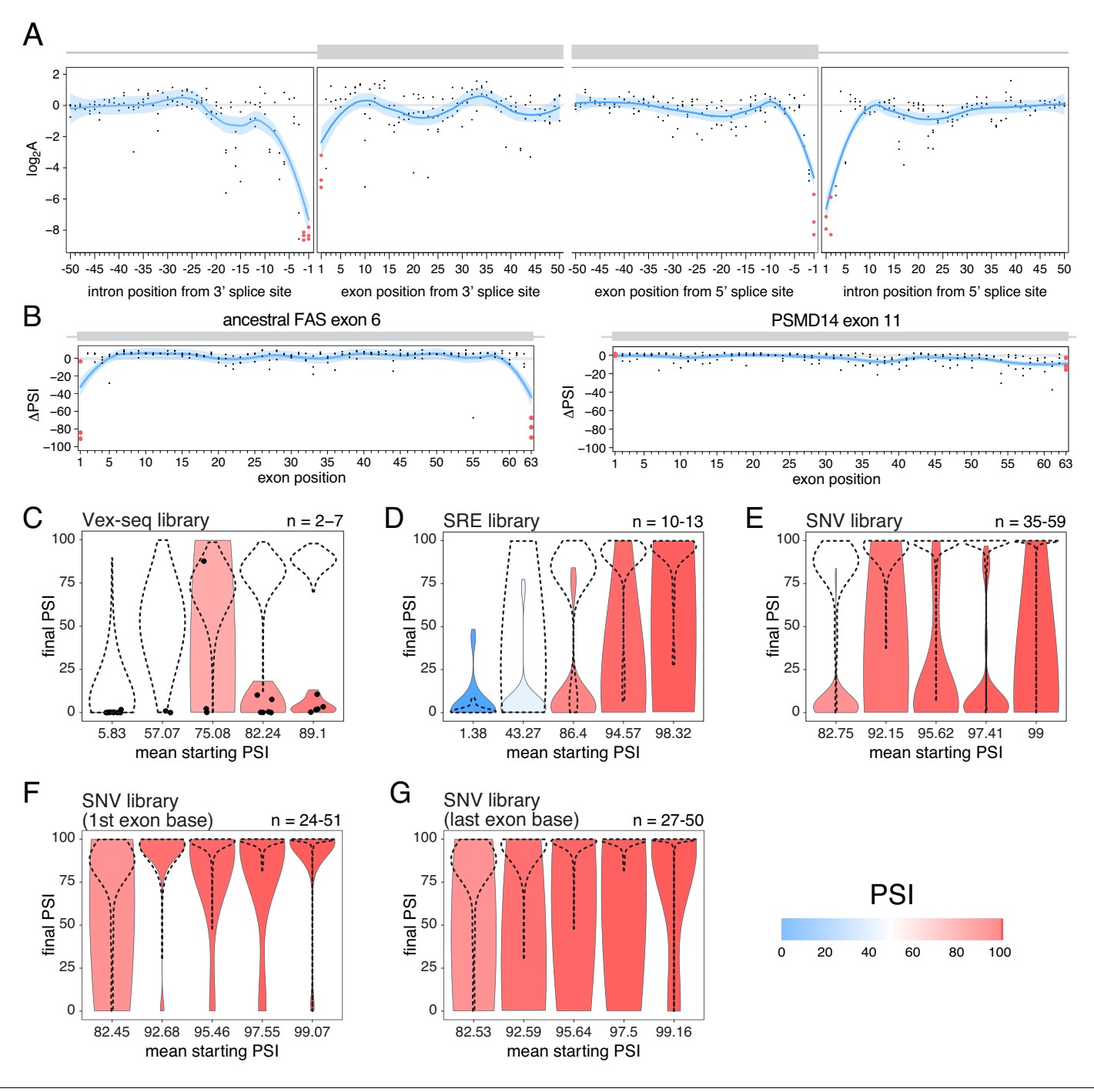

**Figure 8.** The effects of splice site mutations. (A) Position-dependent distribution of biophysical mutation effects in the *RON* exon 11 dataset. Loess curves and their 95% confidence bands are shown in red (all positions) and blue (all positions except for the first and last exonic positions and the invariant intronic dinucleotide positions). (B) Position-dependent distribution of splice-altering effects in the two mutagenesis experiments. Mutations at the first and last exonic positions have a very strong skipping-promoting effect in the ancestral *FAS* exon 6, but not in *PSMD14* exon 11. Loess curves as in A. (C) Distribution of invariant dinucleotide mutation effects in the Vex-seq library, separated into the same bins as *Figure 5B*. The black dashed lines show the distributions of all intronic mutations. The number on the top right represents the number of data points in each bin. Because this number is so low, each individual data point is also shown (represented as a black dot). (D) Distribution of invariant dinucleotide mutation effects in the SRE library, separated into the same bins as *Figure 5C*. The black dashed lines show the distributions of all intronic mutations. (E) Distribution of invariant dinucleotide mutation effects in the SNV library, separated into the same bins as *Figure 5D*. The black dashed lines show the distributions of all intronic mutations. (F) Distribution of first exon position mutation effects in the SNV library, separated into the same bins as *Figure 3D*. The black

*Figure 8 continued on next page*

*Figure 8 continued*
dashed lines show the distributions of all exonic mutations. (G) Distribution of last exon position mutation effects in the SNV library, separated into the same bins as *Figure 3D*. The black dashed lines show the distributions of all exonic mutations.

splicing reaction occurs less efficiently compared to in vivo experiments (*Hicks et al., 2005*), so the mutated exons were never highly included. Additionally, exons were often sensitized to the effects of perturbations by surrounding these exons with the genomic sequences of less included exons (*Liu et al., 1998*; *Schaal and Maniatis, 1999*; *Shiga et al., 1997*; *Tanaka et al., 1994*; *Watakabe et al., 1993*). In agreement with our conclusions, a previous study showed that single and double nucleotide substitutions had little effect on enhancer activity compared with more complex mutations (*Schaal and Maniatis, 1999*). Highly-included exons have been shown to have a higher density of enhancers when compared with alternative exons, which presumably contributes to their high inclusion levels (*Ke et al., 2011*; *Wang et al., 2005*). However, this does not mean that individual point mutations in these elements will produce large changes in inclusion. Indeed, our results revealed that, at the same inclusion level, mutations have a stronger effect in alternative exons than in constitutive exons (*Figure 7*). This may be, in part, precisely due to the higher density of enhancer elements in constitutive exons, which produces sequence redundancy in these exons.

*Zhang et al., 2005* have argued against enhancer redundancy, proposing instead that most or all of the existing enhancers in a given exon are necessary to reach the inclusion levels of a constitutive exon. Indeed, individual cases of single-nucleotide substitutions able to disrupt the splicing of highly-included exons have been reported (*de Boer et al., 2017*; *Zhang et al., 2005*) and also exist in our systematic mutagenesis datasets. However, systematic experiments and genetic analyses argue that these effects are not prevalent. *Figure 9* provides a possible rationale to reconcile these observations. Under most conditions, the high levels of inclusion of constitutive exons create, through the scaling of mutational effects, an apparent redundancy between individual elements, 'locking' the exon in a state of nearly 100% inclusion. When the same exons are included at lower levels, for example due to a change in conditions, the redundancy effect disappears and individual mutations display stronger effects.

The exception to this is mutations that affect splice sites, which can change the inclusion of even highly-included exons (*Figure 6*). Our results reveal that splice site mutations in exons that are mostly skipped can also have an inclusion-promoting effect stronger than that of mutations elsewhere in the transcript. Therefore, the activation of a cryptic splice site (a sequence that resembles a

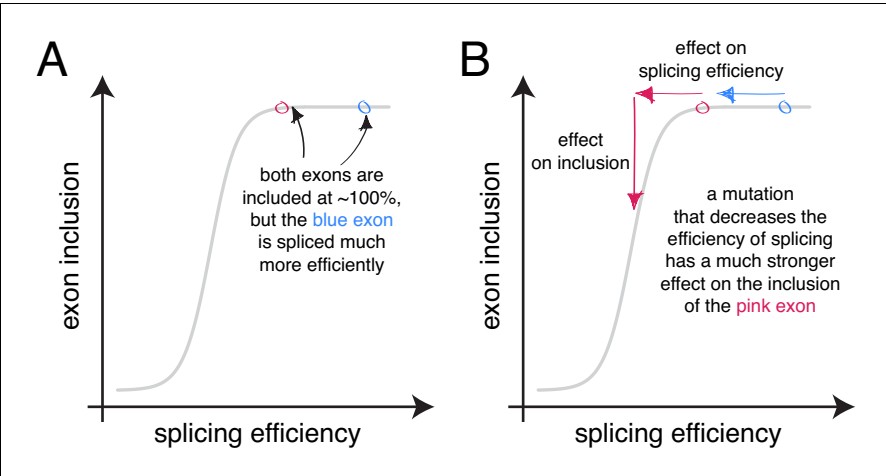

**Figure 9.** The scaling of mutational effects can create what appears to be sequence 'redundancy' as the exon inclusion level approaches 100 or 0%. (**A**) Two exons can be included at nearly 100%, but be spliced with very different efficiencies. (**B**) A mutation that decreases the efficiency of splicing will have a strong effect on the inclusion of the exon with lower splicing efficiency (pink), but almost no effect on the inclusion of the exon with higher splicing efficiency, even though the starting inclusion level was almost the same.

splice site but which is not recognized as such) could be a common mechanism by which distal intronic mutations affect splicing. This is often the case with Alu elements (retrotransposons that make up over 10% of the human genome). When inserted in an antisense orientation, an Alu element contains a number of sequences resembling splice sites (and are therefore cryptic splice sites). Such an Alu element could therefore be thought of as an exon included at 0%. Mutations in these cryptic splice sites have been shown to activate them, resulting in the insertion of a new exon in the mature transcript and causing disease (*Ule, 2013*). Other distal intronic mutations have been shown to cause disease using a similar mechanism. For example, a mutation that strengthens a cryptic splice site was shown to activate a silent exon in the *CAPN3* gene, resulting in limb-girdle muscular dystrophy type 2A (*Blázquez et al., 2013*). Indeed, the activation of cryptic splice sites is one of the most common mechanisms by which distal intronic mutations affect splicing (*Vaz-Drago et al., 2017*).

The small effects of mutations in highly-included exons have implications for the evolution of alternative splicing. The evolution of lower inclusion levels is likely to involve multiple mutations with each of these mutations individually seeming to have no or little effect. This suggests that the evolution of alternative splicing is often likely to initially involve neutral drift until the splicing efficiency of an exon drops enough for mutations to have a noticeable effect on inclusion, allowing skipping-promoting mutations to be selected for or against, if they have a fitness impact. For example, in our recent study a single mutation was never sufficient to decrease the inclusion of the ancestral *FAS* exon six to the inclusion levels of the human exon. Indeed only 8% of double mutants, 11% of triple mutants and 19% of genotypes containing 10 mutations reached the inclusion levels of the human exon (*Baeza-Centurion et al., 2019*).

Sequencing clinical samples frequently returns novel alternative alleles in known or candidate disease genes whose effects on disease are unclear (*ClinGen et al., 2015*). A major challenge in human genetics is to determine whether such variants of unknown significance (VUS) are disease-causing mutations. A second challenge is to understand the mechanisms by which alternative alleles cause disease. Our results suggest that VUS are most likely to alter exon inclusion, and therefore be candidate disease-causing alleles by this mechanism, if they are located in or close to an alternative exon with intermediate levels of inclusion, or in a splice site.

In summary, our results show that very few mutations in both exons and proximal intronic regions typically alter the inclusion of highly-included exons and even fewer impact the inclusion of constitutive exons. Indeed, the distribution of inclusion-altering mutations will be very non-uniform across the genome, focussed around alternatively spliced exons with intermediate levels of inclusion. The exception to this is mutations in splice sites, which can have large effects on inclusion even in highly-included exons. Exons with intermediate inclusion levels make up a minority of expressed exons in any human cell type. Thus, considering the entire human genome, most mutations in exons and introns will actually have negligible effects on exon inclusion. We therefore propose as a general rule to prioritise, as candidate causal disease mechanisms, mutations in canonical splice sites and mutations in and surrounding alternative exons with intermediate levels of inclusion.

## Materials and methods

### Ancestral *FAS* exon six doped library design and synthesis

Library oligonucleotides were designed to include the sequence of the ancestral *FAS* exon 6, doped at each position with 1.2% of each of the three non-reference nucleotides, flanked by invariant sequences corresponding to 22 nucleotides of the 3' end of the human intron 5 and 50 nucleotides of 5' end of the human intron 6:

5'-TGT CCA ATG TTC CAA CCT ACA G**GA TCC AGA TCT AAC TTG CTG TGG TTG TGT CTC CTG CTT CTC CCG ATT CTA GTA ATT GTT TGG G**GT AAG TTC TTG CTT TGT TCA AAC TGC AGA TTG AAA TAA CTT GGG AAG TAG −3'.

Letters in bold indicate the doped exonic sequence. The library was synthesized by Trilink Biotechnologies, and ordered on the 1 μM scale.

## Ancestral *FAS* exon six library amplification and subcloning

Accuprime Pfx (12344024, ThermoFisher Scientific) was used following the manufacturer's instructions to amplify 20 ng of single-stranded library DNA for 25 cycles with the following flanking intronic primers: FAS_i5_GC_F and FAS_i6_GC_R (*Baeza-Centurion et al., 2019*).

The amplified library was then recombined with pCMV FAS wt minigene exon 5-6-7 (*Förch et al., 2000*) using the In-Fusion HD Cloning kit (639649, Clontech) in a 1:8 vector:insert optimized ratio and transformed into Stellar competent cells (636766, Clontech) to maximise the number of individual transformants (800,000 individual clones). The library was then amplified by growing for 18 hr in LB medium containing ampicillin. The final plasmid library was purified using the Quiagen plasmid maxi kit (50912163, Quiagen) and quantified with a NanoDrop spectrophotometer.

## Ancestral *FAS* exon six input library

20 ng of the library was amplified in triplicates using GoTaq flexi DNA polymerase (M7806, Promega) for 25 cycles with three pairs of FAS_i5_BR_F and FAS_i6_BR_R intronic primers (*Baeza-Centurion et al., 2019*) leading to a 135 nucleotide PCR band that was gel-purified and sequenced. Each pair of primers had a distinct 8-mer barcode sequence to discriminate between technical replicates.

## Cell transfection and generation of output libraries

For each of the nine experimental replicates of the ancestral *FAS* exon six library, 10 ng of the library were transfected into 250 000 HEK293 cells in one well of a 6-well plate using Lipofectamine 2000 (11668027, ThermoFisher Scientific) and OPTIMEM I Reduced Serum Medium (31985–047, ThermoFisher Scientific). Six hours post-transfection, the cell culture medium was replaced with DMEM Glutamax (61965059, ThermoFisher Scientific) containing 10% FBS and Pen/Strep antibiotics. 48 hr post-transfection, total RNA was isolated using the automated Maxwell LEV 16 simplyRNA tissue kit (AS1280, Promega). cDNA was synthetized from 500 ng total RNA using Superscript III (18080085, Life Technologies), and amplified with one of the nine pairs of barcoded FAS_e5_BR_F and FAS_e5_BR_R primers (*Baeza-Centurion et al., 2019*) and GoTaq flexi DNA polymerase (M7806, Promega). Each pair of primers had a distinct 8-mer barcode sequence to distinguish the nine experimental replicates. PCR products were run on a 2% agarose gel and the band corresponding in size to the amplification product of exon inclusion was excised, purified using the Quiaquick Gel extraction kit (Quiagen, 50928704) and quantified with a NanoDrop spectrophotometer.

## *PSMD14* exon 11 library

Library oligonucleotides were designed to include the sequence of *PSMD14* exon 11, doped at each position with 1.2% of each of the three non-reference nucleotides, flanked by invariant sequences corresponding to 22 nucleotides of the 3′ end of intron 5 of the *FAS* gene, and 50 nucleotides of 5′ end of intron 6 of the *FAS* gene:

5′-TGT CCA ATG TTC CAA CCT ACA G**GC TGT AGA AGA AGA AGA TAA GAT GAC ACC TGA ACA GCT GGC AAT AAA GAA TGT TGG CAA GCA G**GT AAG TTC TTG CTT TGT TCA AAC TGC AGA TTG AAA TAA CTT GGG AAG TAG −3′.

Letters in bold indicate the doped exonic sequence. The library was synthesized by Trilink Biotechnologies, and ordered on the 1 μM scale.

The remaining procedure was the same as the one described above for the ancestral *FAS* exon six mutagenesis library.

## Sequencing

For the experiment involving the ancestral *FAS* exon 6, equimolar quantities of three independent amplifications of the input library and equimolar quantities of the purified inclusion band (output library) of each of the nine output replicates were pooled and sequenced at the CRG Genomics Core Facility where Illumina Ampliseq PCR-free libraries were prepared and run on a single lane of an Illumina HiSeq3000. In total, over 285 million paired-end reads were obtained. The median sequencing coverage for each single mutant in the input was between 12381 and 26749 reads. In the output, the median sequencing coverage was between 7231 and 28672 reads. The same procedure was followed to sequence the *PSMD14* exon 11 library. For this library over 93 million paired-

end reads were obtained. The median sequencing coverage for each single mutant was between 11887 and 29091 reads in the input and between 15915 and 20088 reads in the output.

## Data processing

Sequencing data from the *FAS* exon six library (*Julien et al., 2016*), the ancestral *FAS* exon six library, and the *PSMD14* exon 11 library were all processed using the *WRAP* module of *DiMSum* v0.1 (https://github.com/lehner-lab/DiMSum) (*Faure et al., 2020*) to calculate the number of counts of every single mutant variant in each input and output. Raw enrichment scores (ES) for all single mutants x in the three libraries were calculated as the ratio between their frequency in the output and the input libraries:

$$ES_{x,raw} = \frac{freq_{x,out}}{freq_{x,in}}$$

Normalised ESs was calculated by dividing the ES of each variant x over the ES of the WT sequence:

$$ES_{x,norm} = \frac{ES_{x,raw}}{ES_{wt,raw}}$$

We used DiMSum's error model (that combines Poisson error in addition to inter-replicate variation, *Faure et al., 2020*) to obtain a standard error for each enrichment score.

## Calculation of PSI values

The PSI values of the WT *FAS* exon six have been previously measured in HEK293 cells to be 49% (*Julien et al., 2016*). In the same cell line, the PSI of the ancestral *FAS* exon six was experimentally determined to be 95% and that of *PSMD14* exon 11 was calculated to be 97% (see 'Experimental evaluation of PSI values' below and *Figure 2—figure supplement 1*).

To calculate the PSI of each sequence variant x, normalised ESs were multiplied by the WT PSI:

$$PSI_x = PSI_{wt} \cdot \frac{ES_{x,norm}}{ES_{wt,norm}} = PSI_{wt} \cdot ES_{x,norm}$$

Some variants had such high enrichment scores their PSI values were estimated to be greater than 100. We therefore use the Bayesian correction procedure described in *Li et al., 2019* to rescale our ESs so they abide by the theoretical maximum and minimum levels of exon inclusion. The minimum possible enrichment score is 0, and the maximum is given by:

$$ES_{norm,max} = \frac{1}{PSI_{wt}} \cdot 100$$

The range of allowed enrichment scores [0, ES$_{norm,max}$] was divided into 1000 evenly spaced numbers k. Next, from a measured ES and its standard deviation, the probability P$_{ES,k}$ of the true ES being k is given by the probability that a value as extreme as k is observed assuming a normal distribution $N\left(ES_x, \varepsilon_{ES_x}^2\right)$. A re-scaled ES was calculated based on the weighted mean of all k numbers, where the weights are the probability of the true ES being each number k:

$$EES_{rescaled} = \frac{\sum_{k=1}^{1000} P_{ES,k} \cdot k}{\sum_{k=1}^{1000} k}$$

Finally, the distribution of rescaled ESs had a mode that was slightly shifted from the mode of the non-rescaled ESs (which falls around the WT score). Since the distribution of single mutant ESs is expected to fall around the WT ES, rescaled ESs were adjusted by subtracting the difference between the mode of their distribution and the WT ES.

## Experimental evaluation of PSI values

Experimentally-determined PSI values for *FAS* exon six were obtained from *Julien et al., 2016*. For the ancestral *FAS* exon six library, we experimentally determined the inclusion of the WT sequence as well as that of a selection of single mutant variants (*Figure 2—figure supplement 1*). To build minigenes containing the correct sequences, oligonucleotides containing the 63-nucleotide-long ancestral exon sequence were ordered from IBA Gmbh on the 0.2 µmol scale. Apart from the exonic sequence, these oligonucleotides are flanked by 22 nucleotides of the 3' end of the human intron 5 and 50 nucleotides of the 5' end of the human intron 6:

5'-TGT CCA ATG TTC CAA CCT ACA G**GA TCC AGA TCT AAC TTG CTG TGG TTG TGT CTC CTG CTT CTC CCG ATT CTA GTA ATT GTT TGG G**GT AAG TTC TTG CTT TGT TCA AAC TGC AGA TTG AAA TAA CTT GGG AAG TAG −3'.

For the *PSMD14* exon 11 library, the oligonucleotides ordered were the following:

5'-TGT CCA ATG TTC CAA CCT ACA G**GC TGT AGA AGA AGA AGA TAA GAT GAC ACC TGA ACA GCT GGC AAT AAA GAA TGT TGG CAA GCA G**GT AAG TTC TTG CTT TGT TCA AAC TGC AGA TTG AAA TAA CTT GGG AAG TAG −3'.

Letters in bold indicate the exonic region. As described previously for the doped libraries, this sample was purified by Reverse Phase HPLC, amplified and recombined with the pCMV FAS wt minigene exon 5-6-7 vector. Mutants were obtained with the Accuprime Pfx DNA polymerase (Thermo scientific, 12344024), following the manufacturer's instructions, and primers were designed using PrimerX (http://www.bioinformatics.org/primerx/). Individual mutants were verified by Sanger sequencing and transfected into HEK293 cells in triplicates to quantify the ratio between exon six inclusion and skipping. For RT-PCR, minigene-specific primers PT1 and PT2 were used (*Baeza-Centurion et al., 2019*) that were designed to avoid amplification of endogenous *FAS* RNAs. RT-PCR products were fractioned by electrophoresis using 6% polyacrylamide gels in 1 x TBE and Sybr safe staining (Thermo scientific, S33102). The bands corresponding to exon inclusion or skipping were quantified using ImageJ 1.47 v (NIH, USA).

## Processing of *WT1* exon five dataset

Pre-processed data from *Ke et al., 2018*, containing variant read counts from four input technical replicates and three output biological replicates, was downloaded from GEO (accession number: GSE105785). The counts from all input replicates were pooled together and normalised enrichment scores for each output replicate were calculated as described above for our *FAS* exon six libraries. 10 different *WT1* exon five libraries (labelled A-J) were generated, each time in the presence of a different hexameric sequence in positions 5–10, effectively resulting in 10 different WT sequences included at different levels:

| Hexamer | PSI of WT sequence with this hexamer |
| --- | --- |
| A | 7% |
| B | 20% |
| C | 65% |
| D | 0.1% |
| E | 3% |
| F | 43% |
| G | 4% |
| H | 74% |
| I | 53% |
| J | 5% |

We only used data from the libraries in which the WT sequence was included at intermediate levels (libraries B, C, F, H and I). The PSI for each single mutant variant was calculated by multiplying its normalised ES times the inclusion levels of its WT. ΔPSI values were calculated by subtracting the PSI of each single mutant from the PSI of the WT sequence.

## Processing of *RON* Exon 11 dataset

We downloaded supplementary data 3 from *Braun et al., 2018*. The fifth sheet (named 'HEK293T average frequency') of this Excel spreadsheet contains the estimated inclusion levels of all single mutants in the *RON* exon 11 dataset. We excluded all mutants classified as insertions ('IN') or deletions ('DEL'). ΔPSI values were calculated by subtracting the PSI of each single mutant from the PSI of the WT sequence. For downstream analyses involving exonic mutations only, we excluded any variants not located in the alternative exon (between positions 298 and 444, both included). For downstream analyses of flanking intronic mutations, we excluded any variants not found in either of the two flanking introns (positions 211 to 297, and 445 to 524).

## Predicting ΔPSI distributions at different starting PSIs

To estimate a distribution of additive mutational effects (that is independent of the starting PSI) in *FAS* exon 6, *RON* exon 11 and *WT1* exon 5, we used *Equation 2* from the main text. To calculate the scaled ΔPSI distribution of each alternative exon dataset at a particular starting PSI, we used *Equation 1*.

To obtain a 95% confidence interval for the scaled distribution of mutational effects (*Figure 2E*), we sampled each of six distributions of additive mutation effects (one for *FAS* exon 6, one for *RON* exon 11 and four for *WT1* exon 5 – *WT1* exon five version B was not included since it systematically predicts ΔPSI values greater than observed, see *Figure 1C*) 2500 times with replacement, and estimated the rescaled PSI distribution from each of these bootstrapped distributions.

We estimated the density of each PSI distribution using the *density* function in R with the *bw* parameter set to 5. Every estimated density distribution was divided into 512 different values according to the PSI. The 95% confidence interval is given by the 2.5th (lower limit) and 97.5th (upper limit) percentile density values in each bin.

## Scaling of DMS datasets

To scale PSI values from the *FAS* exon 6 dataset to 95.1% (the starting PSI of the ancestral *FAS* exon six dataset), to 97.6% (the starting PSI of *PSMD14* exon 11), and 99.4% (the starting PSI of *SMN1* exon 7), we used *Equation 1* above to estimate the additive effects (the change in splicing efficiency A) of all mutations. Next, we used *Equation 2* to scale the distribution of ΔPSI effects to these different starting PSIs.

## *SMN1* exon seven dataset

We downloaded supplementary table 1 from *Souček et al., 2019*. Information about the the inclusion of each mutation is provided in the third sheet (named 'SNV statistics') of this Excel spreadsheet.

## SRE library

Processed data for the SRE library was downloaded from the Kosuri lab's Github repository (https://github.com/KosuriLab/MFASS/blob/master/processed_data/sre/sre_data_clean.txt). Some exons in this library were never mutated. Since we were interested in analysing the effects of mutations on exon inclusion, we excluded these exons from the dataset. To study the effects of exonic mutations, we selected mutations belonging to one of the following categories: 'clst_Ke2011_ESE', 'rmv_Ke2011_ESE', 'clst_Ke2011_ESS', 'rmv_Ke2011_ESS', 'rnd_exon_1nt', 'rnd_exon_2nt', 'rnd_exon_3nt', 'rnd_exon_5nt', 'aggr_exon'. The effect of a mutation was calculated by subtracting the PSI of the exon with the mutation from the PSI without the mutation. The same exon was sometimes mutated more than once. In those cases, we calculated their median effect to ensure that all exons had an equal weight in the final ΔPSI distribution. To study the effects of intronic mutations, we focussed on mutations belonging to one of the following categories: 'clst_Vlkr07_AICS', 'clst_Vlkr07_DICS', 'rmv_Vlkr07_AICS', 'rmv_Vlkr07_DICS', 'rnd_intron_1nt', 'rnd_intron_2nt', 'rnd_intron_3nt', 'rnd_intron_5nt', 'aggr_intron', 'p_aggr_intr'. Mutation effects were calculated as described for exonic mutations. To study the effects of mutations in the invariant intronic dinucleotide positions of the splice sites, we selected mutations belonging to one of the following categories: 'rnd_intron_1nt', 'rnd_intron_2nt', 'rnd_intron_3nt', 'rnd_intron_5nt'; which contained at least

one mutation at the invariant dinucleotide positions. Mutation effects were calculated as described for exonic mutations.

## SNV library

Processed data for the SNV library was downloaded from the Kosuri lab's Github repository (https://github.com/KosuriLab/MFASS/blob/master/processed_data/snv/snv_data_clean.txt). As with the SRE library, we excluded exons from this dataset that were never mutated. To study the effects of exonic mutations, we focussed on mutations belonging to the 'exon' categories, and calculated mutation effects as described above for the SRE library. For intronic mutations, we selected those classified as 'downstr_intron' or 'upstr_intron'. For mutations in the invariant intronic dinucleotides, or in the first and last exonic positions, we selected mutations located at those positions. Mutation effects were calculated as described above.

## Vex-seq library

Processed data from the Vex-seq library were downloaded from the Vex-seq Github repository. Inclusion levels for each variant can be accessed at https://github.com/scottiadamson/Vex-seq/blob/master/processed_files/delta_PSI_values.tsv and the classification of each variant is found at https://github.com/scottiadamson/Vex-seq/blob/master/processed_files/68_97_VEP_multiple.tsv. To study the effects of exonic mutations, we selected mutations belonging to one of the following categories: 'frameshift', 'inframe deletion', 'missense', 'synonymous', 'stop gained', 'stop lost', 'non coding transcript exon'. Mutation-induced ΔPSI values were already provided in the pre-processed data. To study the effects of intronic mutations, we selected mutations classified as 'intron'. To study the effects of mutations in the invariant intron dinucleotides or in the first and last exon positions, we downloaded the file containing information about the position of each variant, which can be found at https://github.com/scottiadamson/Vex-seq/blob/master/processed_files/Vex-seq_positions.tsv, and selected mutations falling at the corresponding positions.

## Distribution of PSI values in the GTEx dataset

To estimate the PSI of all exons in all GTEx samples (*GTEx Consortium, 2017*) from the proportion of reads supporting exon inclusion in the GTEx junction read counts file (*GTEx_Analysis_2016-01-15_v7_STARv2.4.2a_junctions.gct.gz*; available for download at https://www.gtexportal.org/home/datasets), we used the *quantifySplicing* function from the *Psichomics* package in R (*Saraiva-Agostinho and Barbosa-Morais, 2019*). The *minReads* argument was set to 10 (such that a splicing event requires at least 10 reads for it to be quantified) and the *eventType* argument was set to 'SE' (instructing the *quantifySplicing* function to quantify alternative exon events). All estimates were based on the *Psichomics* hg19/GRCh37 alternative splicing annotations.

To generate a genome-wide distribution of exon inclusion levels across all tissues, the mean exon PSI in each tissue across all samples was calculated and visualised as a histogram. The distributions of inclusion values in each tissue were visualised by splitting the data according to tissue. To plot the genome-wide distribution of nucleotide inclusion levels, we first calculated the mean exon PSI in each tissue across all samples. Then, a distribution of inclusion values was generated by sampling each PSI value *n* times, where *n* is the length of each exon. The distributions of nucleotide inclusion values in each tissue were visualised by splitting the data according to tissue.

## ΔPSI effects of common exonic alternative alleles in the GTEx population

The GTEx genotype matrix file (*GTEx_Analysis_2016-01-15_v7_WholeGenomeSeq_635Ind_PASS_AB02_GQ20_HETX_MISS15_PLINKQC.vcf.gz*, provided via dbGap) was filtered for common alternative alleles (minimum allele frequency $\geq$5%) using the BCFTools (http://www.htslib.org) *view* command with the `--min-af` option set to *0.05:minor*. This dataset was further filtered to include only exonic nucleotide substitutions (i.e. exonic insertions and deletions as well as any intronic variants were discarded from our dataset).

The GTEx junction read counts file was processed using the *Psichomics* library in R as described above, and the output of the *quantifySplicing* function was subset to include only samples for which

genotype information was also available. Splicing events that could not be quantified in any of the remaining samples were removed from the dataset.

For each exonic alternative in each exon in each tissue, a linear model was built to predict exon PSI using the number of variant copies as the sole predictor variable. The estimated ΔPSI was given by the slope of the linear model (the amount of exon inclusion changing with each additional copy of the alternative allele) multiplied by two. The model's intercept term was used as an estimate of the exon PSI without the exonic variant (i.e. the 'starting PSI' condition). Although we could simply take the average PSI among all samples lacking a specific variant, some variants were never absent (i.e. all samples had either one or two copies of the variant). Therefore, this method allows us to estimate (extrapolate) what the PSI would be in the absence of the sQTL, even if the data are not available. If the estimated starting PSI was above 100 (or below 0), this variant was removed from our dataset.

This procedure was repeated to study the effects of common intronic alternative allele within 50 nucleotides of an exon, and to study the effects of variants located in the splice sites.

To account for linkage disequilibrium, the final results only consider one alternative allele per splicing event (the one corresponding to the smallest p-value out of all the associations identified with a given splicing event). If an exonic allele was identified as the strongest associated allele for a particular splicing event, this splicing event was not included in the analysis of intronic alternative alleles. If an intronic allele was identified as the strongest allele, the corresponding splicing event was not included in the analysis of exonic alleles.

## Predicted genome-wide δPSI distributions

To predict the distribution of exonic mutation effects, for each exon we sampled from one of our A parameter distributions a number of times proportional to the length of the exon. We then used *Equation 2* in combination with the previously-estimated inclusion levels of the exon to calculate the ΔPSI values for this exon. The combination of ΔPSI predicted for all exons results in a predicted distribution of genome-wide mutation effects. A similar approach was carried out to predict the genome-wide distribution of intronic mutation effects.

## Effects of common alternative alleles in constitutive and alternative exons

Constitutive exons were defined as those with a PSI > 90% in all 30 GTEx tissues (as calculated above in the section titled 'Distribution of PSI values in the GTEx dataset'). Alternative exons were defined as those with a PSI < 90% in at least one tissue. The magnitude of the effects of skipping-promoting mutations (common variants associated with less exon inclusion) for each exon in each tissue were obtained as described above (in the section titled 'ΔPSI effects of common exonic alternative alleles in the GTEx population').

The difference of mutation effects in constitutive versus alternative exons was calculated by building a linear model in which the effect of a mutation is modeled as a function of the starting PSI and the type of exon (constitutive or alternative). The statistical significance for this difference was obtained by performing a t test for the significance of the term in the model corresponding to the type of exon.

This analysis was repeated after using different definitions for constitutive and alternative exons. The definition of constitutive exon was varied by varying the threshold above which it must be included in all tissues to be considered a constitutive exon (an array of 50 numbers, from 50% to 99%). Similarly, the definition of alternative exon was also varied by changing the threshold below which an exon must be included in at least one tissue to be considered alternative (an array of 50 numbers, from 50 to 99%).

## Analysis of ESE and ESS motifs in constitutive and alternative exons

A list of 1182 putative enhancer hexamers and 1090 putative silencer hexamers was obtained from Supplemental Table 1 in *Ke et al., 2011*. To compare ESE density between constitutive and alternative exons, we counted the number of ESE hexamers in each exon normalised by exon length. All exons were divided into 10 groups according to their ESE density, and the proportion of alternative

vs constitutive exons in each group was calculated. The same analysis was performed to study the density of ESS hexamers in constitutive and alternative exons.

To account for the higher PSI of constitutive exons (higher PSI correlates with a higher density of enhancer elements in an exon), this analysis was repeated considering only alternative exons with a PSI > 90% in at least one GTEx tissue. This analysis was further repeated in a tissue-specific manner, only including alternative exons with PSI > 90% in each tissue.

To account for the effect that splice site strength might have on this analysis, we measured the strength of all splice sites in our dataset using MaxEntScan (*Yeo and Burge, 2004*). A splice site was considered to be 'weak' if its MaxEntScan score was among the bottom 20% of scores in our dataset. A splice site was labelled as 'strong' if its MaxEntScan score was among the top 20% in the dataset.

## Analysis of suboptimal ESEs

To compare the density of suboptimal ESE sequences in alternative and constitutive exons, we repeated the analysis described above for ESE motifs, but analysing hexamer motifs that differ from an ESE hexamer by only one nucleotide substitution.

## Data access

Raw sequencing data have been submitted to GEO with accession number GSE151942. All scripts used in this study are available at (https://github.com/lehner-lab/Constitutive_Exons; *Baeza-Centurion, 2020* copy archived at swh:1:rev:b2e04359c9b615ee7a1b826ebaed15f073b87298).

## Acknowledgements

We thank Yamile Márquez and Manuel Irimia for identifying *PSMD14* exon 11 as a constitutive exon whose inclusion levels are conserved across many vertebrate species. Work in B.L.'s is supported by a European Research Council (ERC) Consolidator grant (616434), the Spanish Ministry of Economy and Competitiveness (BFU2017-89488-P and SEV-2012–0208), the AXA Research Fund, the Bettencourt Schueller Foundation, Agència de Gestió d'Ajuts Universitaris i de Recerca (AGAUR, SGR-831), the EMBL Partnership, and the CERCA Program/Generalitat de Catalunya. P.B.-C. was funded in part by a Severo Ochoa PhD fellowship. Work in J.V.'s laboratory is supported by Fundación Botín, Banco de Santander through its Santander Universities Global Division, ERC AdvG 670146, AGAUR, Spanish Ministry of Economy and Competitiveness (BFU 2014–005153, BFU 2017 89308 P, and SEV-2012–0208), the EMBL Partnership, and the CERCA program/Generalitat de Catalunya. The Genotype-Tissue Expression (GTEx) data used for the analyses described in this manuscript were obtained from the GTEx Portal on May 8, 2018 and dbGaP accession number phs000424.v7.p2 on May 8, 2018. The GTEx Project was supported by the Common Fund of the Office of the Director of the NIH and by NCI, NHGRI,NHLBI, NIDA, NIMH, and NINDS.

## Additional information

### Competing interests

Juan Valcárcel: Reviewing editor, *eLife*. The other authors declare that no competing interests exist.

### Funding

| Funder | Grant reference number | Author |
| --- | --- | --- |
| European Research Council | ERC 616434 | Ben Lehner |
| European Research Council | ERC 670146 | Juan Valcárcel |
| Ministerio de Economía y Competitividad | BFU 2017-89488-P | Ben Lehner |
| Ministerio de Economía y Competitividad | BFU 2017-89308-P | Juan Valcárcel |
| Banco Santander | Fundación Botín | Juan Valcárcel |

| Fondation Bettencourt Schueller | Liliane Bettencourt Prize for Life Sciences | Ben Lehner |
| Ministerio de Economía y Competitividad | BFU 2014–005153 | Juan Valcárcel |
| Agència de Gestió d'Ajuts Universitaris i de Recerca | SGR-831 | Ben Lehner |
| Ministerio de Economía y Competitividad | Severo Ochoa PhD fellowship | Pablo Baeza-Centurion |
| Ministerio de Economía y Competitividad | SEV-2012–0208 | Ben Lehner |

The funders had no role in study design, data collection and interpretation, or the decision to submit the work for publication.

## Author contributions

Pablo Baeza-Centurion, Conceptualization, Data curation, Software, Formal analysis, Investigation, Methodology, Writing - original draft, Writing - review and editing; Belén Miñana, Validation, Investigation, Methodology, Writing - review and editing; Juan Valcárcel, Conceptualization, Supervision, Funding acquisition, Project administration, Writing - review and editing; Ben Lehner, Conceptualization, Supervision, Funding acquisition, Writing - original draft, Project administration, Writing - review and editing

## Author ORCIDs

Pablo Baeza-Centurion (iD) https://orcid.org/0000-0002-8141-8075
Belén Miñana (iD) https://orcid.org/0000-0001-7676-5788
Ben Lehner (iD) https://orcid.org/0000-0002-8817-1124

## Decision letter and Author response

Decision letter https://doi.org/10.7554/eLife.59959.sa1
Author response https://doi.org/10.7554/eLife.59959.sa2

# Additional files

## Supplementary files

• Transparent reporting form

## Data availability

Raw sequencing data have been submitted to GEO with accession number GSE151942. All scripts used in this study are available at https://github.com/lehner-lab/Constitutive_Exons (copy archived at https://archive.softwareheritage.org/swh:1:rev:b2e04359c9b615ee7a1b826ebaed15f073b87298/).

The following dataset was generated:

| Author(s) | Year | Dataset title | Dataset URL | Database and Identifier |
| --- | --- | --- | --- | --- |
| Baeza-Centurion P, Miñana B, Valcárcel J, Lehner B | 2020 | Mutations primarily alter the inclusion of alternatively spliced exons | https://www.ncbi.nlm.nih.gov/geo/query/acc.cgi?acc=GSE151942 | NCBI Gene Expression Omnibus, GSE151942 |

The following previously published datasets were used:

| Author(s) | Year | Dataset title | Dataset URL | Database and Identifier |
| --- | --- | --- | --- | --- |
| Julien P, Miñana B, Baeza-Centurion P, Valcárcel J, Lehner B | 2016 | The Complete Local Genotype-Phenotype Landscape for the Alternative Splicing of a Human Exon | https://www.ebi.ac.uk/ena/data/view/PRJEB13140 | European Nucleotide Archive (accession code PRJEB13140), PRJEB13140 |

| Ke S, Anquetil V, Zamalloa JR, Maity A | 2018 | Saturation mutagenesis reveals manifold determinants of exon definition | https://www.ncbi.nlm.nih.gov/geo/query/acc.cgi?acc=GSE105785 | NCBI Gene Expression Omnibus, GSE105785 |
|---|---|---|---|---|
| Cheung R, Insigne KD, Yao D, Burghard CP, Wang J, Hsiao Y-H, Jones EM, Goodman DB, Xiao X, Kosuri S | 2019 | A Multiplexed Assay for Exon Recognition Reveals that an Unappreciated Fraction of Rare Genetic Variants Cause Large-Effect Splicing Disruptions (SRE Library) | https://github.com/KosuriLab/MFASS/blob/master/processed_data/sre/sre_data_clean.txt | Github, KosuriLab/MFASS |
| Cheung R, Insigne KD, Yao D, Burghard CP, Wang J, Hsiao Y-H, Jones EM, Goodman DB, Xiao X, Kosuri S | 2019 | A Multiplexed Assay for Exon Recognition Reveals that an Unappreciated Fraction of Rare Genetic Variants Cause Large-Effect Splicing Disruptions (SNV Library) | https://github.com/KosuriLab/MFASS/blob/master/processed_data/snv/snv_data_clean.txt | Github, KosuriLab/MFASS |
| Adamson SI, Zhan L, Graveley BR | 2018 | Vex-seq: high-throughput identification of the impact of genetic variation on pre-mRNA splicing efficiency | https://github.com/scottiadamson/Vex-seq/blob/master/processed_files/delta_PSI_values.tsv | Github, scottiadamson/Vex-seq |
| GTEx Consortium | 2017 | Genetic effects on gene expression across human tissues (junction read counts file GTEx_Analysis_2016-01-15_v7_STARv2.4.2a_junctions.gct.gz) | https://www.gtexportal.org/home/datasets | GTEx Portal, v7 |
| GTEx Consortium | 2017 | Genetic effects on gene expression across human tissues (genotype matrix file GTEx_Analysis_2016-01-15_v7_WholeGenomeSeq_635Ind_PASS_AB02_GQ20_HETX_MISS15_PLINKQC.vcf.gz) | https://www.ncbi.nlm.nih.gov/projects/gap/cgi-bin/study.cgi?study_id=phs000424.v7.p2 | NCBI database of Genotypes and Phenotypes (dbGaP), phs000424.v7.p2 |

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
