## [Decision Letter]

**Acceptance summary:**

Previous work led to a general expectation that genetic variants commonly have large effects on exon inclusion levels and that these will often explain disease risk or trait variation. This manuscript corrects this more general view and, importantly, indicates which genetic variants are most/least likely to alter disease risk. The study resolves this issue at depth (across all PSI values) using extensive genetic data sets.

**Decision letter after peer review:**

Thank you for submitting your article "Mutations primarily alter the inclusion of alternatively spliced exons" for consideration by *eLife*. Your article has been reviewed by three peer reviewers, and the evaluation has been overseen by a Reviewing Editor and Patricia Wittkopp as the Senior Editor. The following individual involved in review of your submission has agreed to reveal their identity: Nuno L Barbosa-Morais (Reviewer #1).

The reviewers have discussed the reviews with one another and the Reviewing Editor has drafted this decision to help you prepare a revised submission.

Summary:

This is an extensive and generally carefully executed study of existing mutagenesis data, human data from GTEx, and some novel mutagenesis data. Results represent the most in-depth, genome-wide comparative examination of the impact of sequence variation on splicing of exons over the spectrum of PSI levels. The authors show that exons of intermediate inclusion levels are much more easily affected by genetic effects on splicing, with high included/excluded exons being much more robust to genetic effects. This represents an important and potentially clinically useful observation. Furthermore, via an analysis of exonic splicing enhancers (ESEs), they add some mechanistic interpretability to this general finding.

Essential revisions:

1) The GTEx analysis needs to account for linkage disequilibrium in order to account for non-independent tests. The approach using a simple effect size cutoff ("only 14% of which are associated with changes in inclusion greater than 10%") is insufficient because it leads to issues with multi-counting data and some difficulties in biological interpretation. Including each variant-exon pair in the downstream analysis pools data from different tissues and introduces massive non-independence in the data. The data should be filtered to have each exon represented only once by either: (a) performing sQTL mapping and then choosing the variant with the best p-value to represent a given variant association to splicing (or fine-mapping analysis), or (b) simply merging the intronic/exonic analyses (or only analyze e.g. exons) because – as you argue – the GTEx variants that you analyze are not necessarily the causal exonic or intronic variants that you seek to study. Our preference is (a).

2) Your study would be more insightful if you could investigate, at greater detail, determinants underlying the effects of mutations that have the greatest impact on PSIs of constitutive/highly included exons. In addition to examining relative splice site strength, such an analysis could also take into account the frequency at which mutations result in the creation of predicted exonic splicing silencers. Your ESE analysis should take account of both splice site strength and ESE frequency/content and go beyond previous analyses, e.g. Fairbrother, Burge and colleagues, Science 2002, because of the richer data set. For example, what is the relative impact of mutations on constitutive exons with weaker vs. stronger 5' and or 3' splice sites? Does a higher frequency of ESEs in constitutive exons confer robustness to effects of mutations even when the flanking splice sites are relatively weak?

3) The authors refer to the potential significance of the results when interpreting variant effects in disease studies, yet should provide an analysis that does this. Using a resource such as ClinVar, there is an opportunity to determine whether otherwise similarly annotated variants (e.g. synonymous variants) in exons of intermediate inclusion levels are more likely to be implicated in disease.

---

## [Author Response]

Essential revisions:1) The GTEx analysis needs to account for linkage disequilibrium in order to account for non-independent tests. The approach using a simple effect size cutoff ("only 14% of which are associated with changes in inclusion greater than 10%") is insufficient because it leads to issues with multi-counting data and some difficulties in biological interpretation. Including each variant-exon pair in the downstream analysis pools data from different tissues and introduces massive non-independence in the data. The data should be filtered to have each exon represented only once by either: (a) performing sQTL mapping and then choosing the variant with the best p-value to represent a given variant association to splicing (or fine-mapping analysis), or (b) simply merging the intronic/exonic analyses (or only analyze e.g. exons) because – as you argue – the GTEx variants that you analyze are not necessarily the causal exonic or intronic variants that you seek to study. Our preference is (a).

We thank the reviewers for pointing this out, and we have corrected our analyses accordingly. To account for non-independent tests, we have now repeated the GTEx analysis in a tissue-specific manner. To account for linkage disequilibrium, for each splicing decision we now consider only the strongest associated variant (the one with the smallest p-value). If the strongest associated variant is inside the exon, this splicing event was not included in the intronic analysis. Conversely, if the strongest associated variant is located in the surrounding introns, we did not include this splicing event in the exon analysis. The new results, shown in Figures 4 and 7, confirm that common alternative alleles have stronger effects on the inclusion of exons that display intermediate exon inclusion levels.

2) Your study would be more insightful if you could investigate, at greater detail, determinants underlying the effects of mutations that have the greatest impact on PSIs of constitutive/highly included exons. In addition to examining relative splice site strength, such an analysis could also take into account the frequency at which mutations result in the creation of predicted exonic splicing silencers. Your ESE analysis should take account of both splice site strength and ESE frequency/content and go beyond previous analyses, e.g. Fairbrother, Burge and colleagues, Science 2002, because of the richer data set. For example, what is the relative impact of mutations on constitutive exons with weaker vs. stronger 5' and or 3' splice sites? Does a higher frequency of ESEs in constitutive exons confer robustness to effects of mutations even when the flanking splice sites are relatively weak?

We thank the reviewers for their suggestions. We have now included an analysis of predicted exonic splicing silencers (ESS) including the density of ESS sequences as well as suboptimal ESS sequences (sequences that are one Hamming distance away from an ESS sequence) in alternative and constitutive exons (Figure 7—figure supplement 7). The results argue that, in contrast to ESEs, ESSs are enriched in alternatively spliced exons compared to constitutive exons, although ESSs are not found in alternative exons which are highly included in at least one tissue. In addition to this, we now analyze the relationship between splice site strength and ESE density (Figure 7—figure supplement 6), ESS density (Figure 7—figure supplement 7) and ESE robustness (Figure 7—figure supplement 12) and find that the conclusions about differential enrichment of ESEs and ESSs hold true regardless of splice site strength, although as expected, constitutive exons are enriched in the subset of the data containing the strongest splice sites, and alternative exons in the subset of the data containing the weakest splice sites. Finally, we now study how often ESS sequences are created after disrupting an already existing ESE (Figure 7—figure supplement 14). Our conclusion is that there is no correlation between the number of ESS sites created and the classification of exons into constitutive and alternative exons.

3) The authors refer to the potential significance of the results when interpreting variant effects in disease studies, yet should provide an analysis that does this. Using a resource such as ClinVar, there is an opportunity to determine whether otherwise similarly annotated variants (e.g. synonymous variants) in exons of intermediate inclusion levels are more likely to be implicated in disease.

We thank the reviewers for their suggestion. Unfortunately, there are only 33 synonymous mutations classified as ‘pathological’ in ClinVar, located in 15 different exons, which are insufficient to provide the statistical power necessary to perform such an analysis.